# TRAINING-FREE CAMERA CONTROL FOR VIDEO GENERATION

**Chen Hou, Zhibo Chen**
University of Science and Technology of China
{houchen@mail.,chenzhibo@}ustc.edu.cn

## ABSTRACT

We propose a training-free and robust solution to offer camera movement control for off-the-shelf video diffusion models. Unlike previous work, our method does not require any supervised finetuning on camera-annotated datasets or self-supervised training via data augmentation. Instead, it is plug-and-play with most pretrained video diffusion models and can generate camera-controllable videos with a single image or text prompt as input. The inspiration for our work comes from the layout prior that intermediate latents encode for the generated results, thus rearranging noisy pixels in them will cause the output content to relocate as well. As camera moving could also be seen as a type of pixel rearrangement caused by perspective change, videos can be reorganized following specific camera motion if their noisy latents change accordingly. Building on this, we propose **CamTrol**, which enables robust camera control for video diffusion models. It is achieved by a two-stage process. First, we model image layout rearrangement through explicit camera movement in 3D point cloud space. Second, we generate videos with camera motion by leveraging the layout prior of noisy latents formed by a series of rearranged images. Extensive experiments have demonstrated its superior performance in both video generation and camera motion alignment compared with other finetuned methods. Furthermore, we show the capability of CamTrol to generalize to various base models, as well as its impressive applications in scalable motion control, dealing with complicated trajectories and unsupervised 3D video generation. Videos available at **https://lifedecoder.github.io/CamTrol/**.

## 1 INTRODUCTION

As a more appealing and content-richer modality, videos differ from images by including an extra temporal dimension. This temporal aspect provides increased versatility for depicting diverse and dynamic movements, which can be decomposed into object motion, background transitions, and perspective changes. Recent years have witnessed the rapid development and splendid breakthroughs of video generation with text prompts or images as input instructions (Li et al., 2023; Hong et al., 2022; Ho et al., 2022; Luo et al., 2023; Zeng et al., 2024; Blattmann et al., 2023a; Brooks et al., 2024; Ge et al., 2023; Fei et al., 2023), demonstrating the immense potential of diffusion models to synthesize realistic videos. While these video generation models have made progress in generating videos with highly dynamic objects and backgrounds (Zeng et al., 2024; Blattmann et al., 2023a; Li et al., 2023), most of them fail to provide camera control for the generated videos.

The difficulty of controlling camera motion in videos arises primarily from two aspects. The initial challenge lies in the inadequacy of annotated data, as most video annotations lack detailed descriptions, particularly about the camera movements. As a result, video generation models trained on these data often fail to interpret text prompts related to camera motions and generate correct outputs. One solution to mitigate the data insufficiency problem is to mimic videos with camera movements through simple data augmentation (Yang et al., 2024a). However, these methods can only handle simple camera motions like *zoom* or *truck*, and have trouble dealing with more complicated ones. The second challenge is the extra finetuning required for camera control and its inherent limitations. As camera trajectories could be sophisticated, they sometimes cannot be accurately elaborated using naive text prompts alone. Common solutions (Wang et al., 2023; He et al., 2024) proposed

to embed camera parameters into diffusion models through learnable encoders and perform extensive finetuning on large-scale datasets with detailed camera trajectories. However, such datasets as RealEstate10k (Zhou et al., 2018) and MVImageNet (Yu et al., 2023) are highly limited in scale and diversity due to the difficulty associated with data collection; in this way, these finetuning methods demand substantial resources but exhibit limited generalizability to other types of scenes. *Lack of annotations and the constraints of finetuning make camera control a challenging task in video generations.*

In this work, we attempt to address these issues through a training-free solution to offer camera control for off-the-shelf video diffusion models. We begin by introducing two core observations underpinning the idea that video diffusion models can achieve camera movement control in a *training-free* manner. First, we find that base video models could produce results with rough camera moves by integrating specific camera-related text into input prompts, such as *camera zooms in* or *camera pans right*. This simple implementation, though not very accurate and always leading to static or wrong motions, shows the natural prior knowledge learned by pretrained models about following different camera trajectories. The other observation is the effectiveness of video models in adapting to 3D generation tasks. Recent works (Voleti et al., 2024; Melas-Kyriazi et al., 2024; Shi et al., 2023) find that leveraging pretrained video models as initialization helps drastically improve the performance of multi-view generations, demonstrating their strong ability to handle perspective change. The two crucial observations reveal the hidden power of video models for camera motion control. Therefore, we seek to find a way to activate this innate ability, as it already exists in the model itself.

We propose **CamTrol**, which offers camera control for off-the-shelf video diffusion models in a training-free but robust manner. CamTrol is inspired by the layout prior that noisy latents possess for the generation results: As pixels in noisy latents change their positions, corresponding rearrangement will also occur in the output, leading to layout modification. Considering camera moves can also be seen as a type of layout rearrangement, this prior can serve as an effective hint, providing the video model with information about specific camera motions. Specifically, CamTrol consists of a two-stage procedure. In stage I, explicit camera movements are modeled in 3D point cloud representation and produce a series of rendered images indicating specific camera movements. In stage II, the layout prior of noisy latents is utilized to guide video generations with camera movements. Compared with previous works, CamTrol requires no additional finetuning on camera-annotated datasets, nor does it need self-supervised training based on data augmentation. Extensive experiments have demonstrated its superior performance in both video generation quality and camera motion alignment against other finetuned methods. Furthermore, we show CamTrol's capability to generalize to various base models, as well as its impressive applications in scalable motion control, dealing with complicated trajectories and unsupervised 3D video generation.

## 2 RELATED WORK

**Camera Control for Video Generation**    While methods aiming to control video foundation models continue to emerge (Ma et al., 2024; Liu et al., 2023; Feng et al., 2023), there are few works exploring how to manipulate the camera motion of generated videos. Earlier works (Hao et al., 2018) control motion trajectory via warping images through densified sparse flow and pixel fusion. Similar ideas also appear later in Chen et al. (2023) and Yin et al. (2023). Apart from utilizing optical flow, two main techniques for implementing video camera control are self-supervised augmentation or additional finetuning. Yang et al. (2024a) disentangles object motion with camera movement and incorporates extra layers to embed camera motions, where the model is trained in a self-supervised manner by augmenting input videos to simulate simple camera movements. He et al. (2024) and Wang et al. (2023) train additional camera encoders and integrate the outputs into the temporal attention layers of U-Net. Guo et al. (2023) learns new motion patterns via LoRA (Hu et al., 2021) and finetuning with multiple reference videos.

**Noise Prior of Latents in Diffusion Model**    One of the most natural advantages of diffusion models comes from its pixel-wise noisy latents formed during the denoising process. These latents have a strong causal effect on the output and directly determine what the result looks like, meanwhile having robust error resilience as they are perturbed by Gaussian noise across different scales. Numerous works have exploited the convenience of this noise prior to attain controllable generation, such as image-to-image translation (Meng et al., 2021), pixel-level manipulation (Nichol et al., 2022), im-

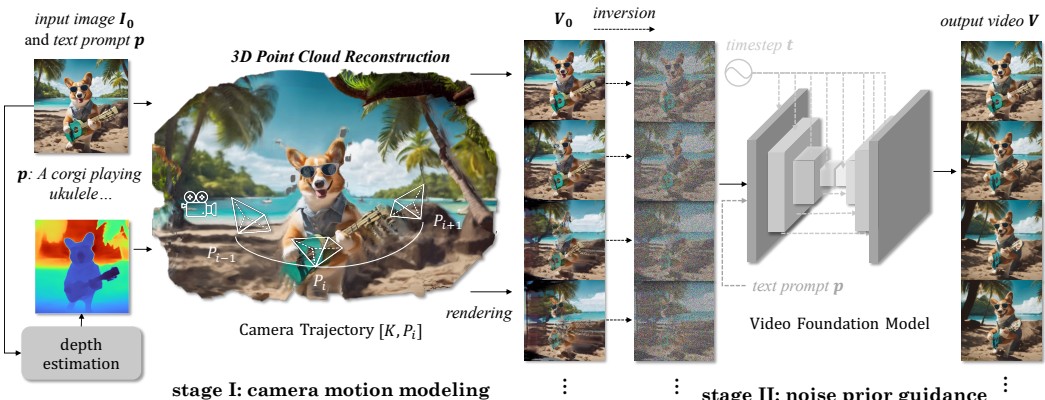

Figure 1: **Pipeline of CamTrol**. In stage I, camera movements are modeled through explicit 3D point cloud. In stage II, the layout prior of noisy latents is utilized to guide video generation.

age inpainting (Lugmayr et al., 2022) and semantic editing (Choi et al., 2021; Hou et al., 2024). Recent research has shown that, although sampled from a Gaussian distribution, the initial noise of the diffusion process still has a significant influence on the layout of generated content (Mao et al., 2023). In other works, noise prior is used to guarantee temporal consistency between video frames (Luo et al., 2023), or to trade off between fidelity and diversity of image editing (Kim et al., 2022).

**Video Model for 3D Generation**    Similar to how most video generation models use the groundwork laid by image foundation models (Blattmann et al., 2023b; Esser et al., 2023; Singer et al., 2022; Wu et al., 2023), the training of 3D generation models also relies heavily on pretrained 2D video models (Voleti et al., 2024; Melas-Kyriazi et al., 2024; Shi et al., 2023; Chen et al., 2024; Han et al., 2024). These methods either finetune with rendered videos directly (Blattmann et al., 2023a; Chen et al., 2024; Melas-Kyriazi et al., 2024; Han et al., 2024), or add camera embedding as an extra condition for each view (Voleti et al., 2024; Shi et al., 2023). Video foundation models have been shown to be particularly beneficial in generating consistent multi-view rendering of 3D objects, demonstrating their inherent and abundant prior knowledge for handling camera pose change.

## 3    TRAINING-FREE CAMERA CONTROL FOR VIDEO GENERATION

CamTrol takes two stages to activate the innate camera control ability hidden in foundation models. In Sec. 3.1, we will describe how to model explicit camera movement for video generation. In Sec. 3.2, we will elaborate on video motion control with the guidance of noise layout prior.

### 3.1    CAMERA MOTION MODELING

To activate pretrained video diffusion models' ability to deal with camera perspective changes, hints of camera motion should be injected into the diffusion model in a proper way. While simply concatenating camera trajectories with text prompts is incomprehensible to the original model, previous works (Wang et al., 2023; He et al., 2024) introduce additional embedders to encode camera parameters and finetune with limited annotated data (Zhou et al., 2018; Yu et al., 2023), which are data-hungry yet lack generalization ability. Other methods (Yang et al., 2024a) construct camera motions by self-supervised augmentations, but can only handle a few easy camera controls. Thus, we seek a more efficient and robust way to guide the model towards being camera-controllable.

Considering that video perspective changes are originally caused by camera movements in 3D space, we leverage 3D representation to provide generation models with explicit motion hints. Specifically, we choose point cloud as the intermediate representation, in which we can easily manipulate camera poses to simulate diverse movements. An additional benefit that point cloud brings is its data efficiency: By utilizing inpainting techniques, only one single input image is required for the entire point cloud reconstruction. This eliminates the need of large-scale finetuning.

**Point Cloud Initialization**    We start by lifting pixels in the input image plane into 3D point cloud space. In practice, the input image can be either user-defined or created by image generators like Stable Diffusion (Rombach et al., 2022). Given an input image $\mathbf{I}_0 \in \mathbb{R}^{3 \times H \times W}$, we first estimate its depth map $\mathbf{D}_0$ using the off-the-shelf monocular depth estimator ZoeDepth (Bhat et al., 2023). By combining the image and its depth map, we initialize the point cloud $\mathcal{P}_0$ as:

$$\mathcal{P}_0 = \phi([\mathbf{I}_0, \mathbf{D}_0], \mathbf{K}, \mathbf{P}_0), \tag{1}$$

where $\phi$ denotes the mapping function from RGBD to 3D point cloud, $\mathbf{K}$ and $\mathbf{P}_0$ represent camera's intrinsic and extrinsic matrices set by convention (Chung et al., 2023) as they're usually intractable.

**Camera Trajectories**    To get consistent images from multiple views, we model the camera motion as a trajectory of extrinsic matrices $\{\mathbf{P}_1, ..., \mathbf{P}_{N-1}\}$, each including a rotation matrix and a translation matrix representing camera pose and position. At each step $i$, we project the point cloud back to the camera plane using $\psi$ and get a rendered image with perspective change: $\mathbf{I}_i = \psi(\mathcal{P}_i, \mathbf{K}, \mathbf{P}_i)$. By computing the extrinsic matrices of different movements, we obtain a series of camera motions including zoom, tilt, pan, pedestal, truck, roll, and rotation, enabling flexible camera movements. Detailed definitions of these movements are provided in Appendix B. Basic trajectories are combined to produce hybrid camera movements, adding cinematic appeal to the generated videos. Furthermore, benefiting from explicit camera motion modeling, our method can support trajectories with precise extrinsics, which means it can generate videos with arbitrarily complex camera motions.

**Multi-view Consistency**    When perspective changes, holes can appear as some areas are unoccupied within the point cloud. To obtain more reasonable results, we employ image inpainting (Rombach et al., 2022) to fill the gaps in new renderings, with a mask distinguishing the known points from the nonexistent ones. This operation ensures coherence between the known and novel views in 2D space. After inpainting, the image is lifted back into 3D space and gradually completes the whole point cloud. During this process, points between adjacent views may become misaligned since the depth estimator only estimates relative depth, leading to inconsistencies in 3D point cloud and rendered images. To avoid this situation, we adopt depth coefficient optimization (Chung et al., 2023) at each step of the camera movement, formed as:

$$d_i = \underset{d}{\arg\min} \left( \sum_M \left\| \phi([\tilde{\mathbf{I}}_i, d\tilde{\mathbf{D}}_i], \mathbf{K}, \mathbf{P}_i) - \mathcal{P}_{i-1} \right\| \right), \tag{2}$$

where $\tilde{\mathbf{I}}_i$ and $\tilde{\mathbf{D}}_i$ refer to the inpainted image and the corresponding depth map, respectively, $d_i$ denotes the depth coefficient to be optimized, and $M$ refers to the overlapping region between $\mathcal{P}_i$ and $\mathcal{P}_{i-1}$, as other areas are not shared for calculating $\ell_1$ loss.

Thus, we get a set of images that correspond to the input and indicate specific camera movements:

$$\{\mathbf{I}_0, ..., \mathbf{I}_{N-1}\} = \{\psi(\mathcal{P}_i, \mathbf{K}, \mathbf{P}_i) | i \in [0, N-1]\}. \tag{3}$$

## 3.2    LAYOUT PRIOR OF NOISE

With camera motion modeling, we obtain a sequence $\mathbf{V}_0 = \{\mathbf{I}_0, ..., \mathbf{I}_{N-1}\} \in \mathbb{R}^{N \times 3 \times H \times W}$ of rendered images adhering to a specific camera trajectory. Note that the quality of rendered images is not perfect since a single input image only leads to sparse point cloud reconstruction. Moreover, these renderings are static, thus they cannot be used directly as video frames. To form an ideal video, we need to find a way that satisfies three requirements: 1) camera motions should be maintained; 2) the video should be enriched with more dynamics; and 3) quality imperfection should be compensated.

**Camera Motion Inversion**    Recent work on diffusion models has demonstrated the strong controllability of their noisy latents (Meng et al., 2021; Mao et al., 2023), the causality and error-resilience they possess with respect to the final output make them a convenient yet powerful tool for controllable generation in diffusion models. Particularly for initial noise, even when sampled from a Gaussian distribution, it still has significant influence on the layout of the generated image (Mao et al., 2023). For instance, if all pixels in initial noise shift to the right by a certain distance, it is

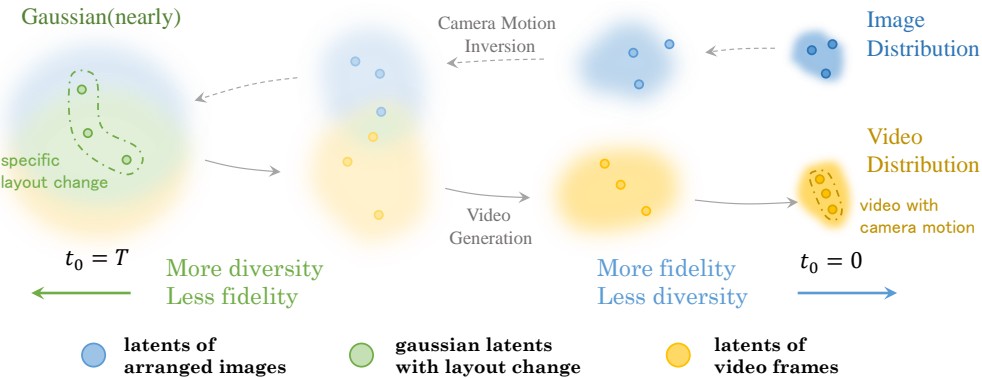

Figure 2: **Transition of samples between two distinct distributions.** As the layout-arranged images are inverted by adding random noise, the distribution of their noisy latents will gradually converge to that of their video counterpart (green area), eventually forming a nearly Gaussian latent with specific layout change. This information is then inherited during the generation process. The inversion step determines the trade-off between video diversity and motion fidelity.

likely that the generated output reflects a similar shift. This reminds us that the impact of camera movement on images can also be interpreted as a type of layout rearrangement, where pixels change their positions due to viewpoint change. In a similar way, videos can be reorganized following camera motion if their noisy latents change accordingly.

Inspired by this, we first construct a series of noisy latents indicating specific camera movements. It can be done intuitively by employing diffusion's inversion process on the rendered image sequence $\mathbf{V}_0$. Latent at timestep $t_0$ can be calculated as follows, where $\bar{\alpha}_t$ is the variance of the scheduler:

$$\mathbf{V}_{t_0} = \sqrt{\bar{\alpha}_{t_0}}\mathbf{V}_0 + \sqrt{1 - \bar{\alpha}_{t_0}}\epsilon, \quad \epsilon \sim \mathcal{N}(\mathbf{0}, \mathbf{I}), \tag{4}$$

Because the rendered images $\mathbf{V}_0$ share common pixels in certain regions, their latents are also correlated to each other in a way reflecting pixel movement. Moreover, by introducing random noise, blank spaces and flawed regions in $\mathbf{V}_0$ can be adaptively completed, providing the video model with more possibilities to generate and correct them.

**Video Generation** After camera motion inversion, noisy latents representing camera movements are then passed through the backward process of the video diffusion model, leveraging their layout controllability to guide video generation. Using prior knowledge of the base model, the generation process also infuses the video with rational dynamic information. In this way, explicit camera movements are injected into the video diffusion model in an appropriate, training-free fashion. Starting from noisy motion latents at timestep $t_0$, the generation step can be represented as:

$$\hat{\mathbf{V}}_{t-1} = \sqrt{\alpha_{t-1}}\left(\frac{\mathbf{V}_t - \sqrt{1 - \alpha_t}\epsilon_\theta^{(t)}(\mathbf{V}_t)}{\sqrt{\alpha_t}}\right) + \sqrt{1 - \alpha_{t-1} - \sigma_t^2}\epsilon_\theta^{(t)}(\mathbf{V}_t) + \sigma_t\epsilon, \quad t \in [1, t_0]. \tag{5}$$

Here $\epsilon_\theta$ denotes the video model for noise prediction and $\sigma_t$ determines whether the denoising process is deterministic or probabilistic. We set $\sigma = 1$ to encourage diversity in generation process.

**Trade-off Between Fidelity and Diversity** Leveraging noise prior guidance in diffusion models could lead to a trade-off between generation fidelity and diversity (Meng et al., 2021; Hou et al., 2024), where results that are more faithful to the guidance tend to degrade in generation quality. In this task, similar circumstances also occur, as the model is required to be guided by some imperfect renderings while generating a reasonable video. The key factor in balancing the trade-off problem lies in the choice of $t_0$. When a smaller $t_0$ is applied, the generated output bears more resemblance to original guidance $\mathbf{V}_0$, but lacks rationality and dynamics to be an appealing video. Instead, a larger $t_0$ leads to better-generated videos but makes them less aligned with the desired camera motion. Empirically, we find that a smaller $t_0$ works better for motions with moderate intensity, and for those with relatively drastic movements, a larger $t_0$ shows preferable performance. The process of stage II is illustrated in Fig. 2.

Table 1: **Quantitative comparisons.** Our method attains comparable performance with finetuned methods in both video generation quality and camera motion alignment.

| Method | Video Quality | | | | Motion Accuracy | | |
|---|---|---|---|---|---|---|---|
| | FVD ↓ | FID ↓ | IS ↑ | CLIP-SIM ↑ | ATE ↓ | RPE-T ↓ | RPE-R ↓ |
| *SVD* | 1107.93 | 68.51 | 7.21 | 0.3095 | 4.23 | 1.79 | 0.021 |
| MotionCtrl+SVD | 810.59 | 69.03 | **7.17** | 0.3076 | 4.19 | **1.07** | 0.012 |
| CameraCtrl+SVD | 951.80 | **67.59** | 6.82 | **0.3138** | 4.22 | 1.17 | 0.013 |
| **CamTrol+SVD** | **778.46** | 68.06 | 7.05 | 0.3110 | **4.17** | **1.07** | **0.010** |
| *Reference* | - | - | - | - | *3.60* | *0.89* | *0.008* |

*A bedroom with a large bed and a television*

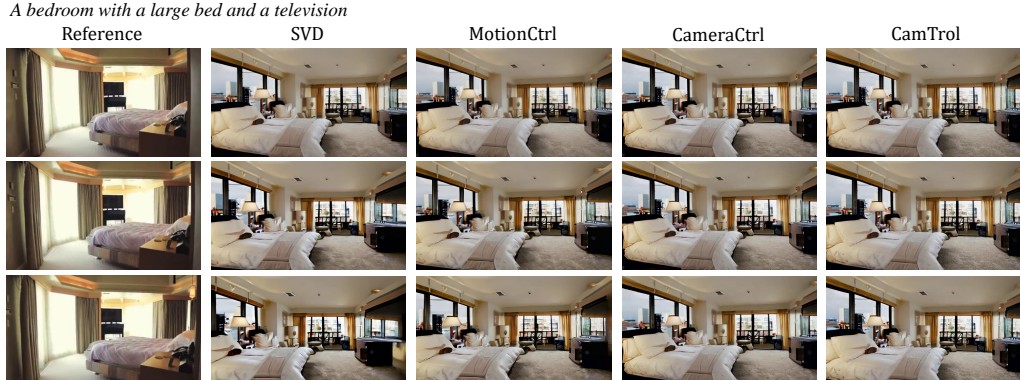

Figure 3: **Qualitative comparisons with finetuned methods.** CamTrol's outputs align well with complex trajectories from the reference video, while others fail to capture subtle changes in camera pose. We provide videos in the supplementary materials for clearer comparison.

## 4 EXPERIMENTS

### 4.1 EXPERIMENTAL SETTINGS

**Implementation Details** We compare our method with state-of-the-art works including MotionCtrl (Wang et al., 2023) and CameraCtrl (He et al., 2024). To ensure a fair comparison, we employ SVD (Blattmann et al., 2023a) as base model for all methods. SVD is originally trained at a resolution of $576 \times 1024$, but SVD-based CameraCtrl only supports $320 \times 576$. Since changing the original resolution results in suboptimal generation quality, we use $576 \times 1024$ for MotionCtrl and CamTrol, then resize the outputs to $320 \times 576$ to calculate metrics. For all methods, the number of frames and the decoding size of SVD are set to 14. We use 25 steps for both the inversion and generation processes.

**Evaluation Details** In the quantitative evaluation, FVD (Unterthiner et al., 2018), FID (Heusel et al., 2017) and IS (Saito et al., 2020) are used to assess video generation quality, and CLIPSIM (Wu et al., 2021) quantifies the similarity between the generated video and the input prompt. For camera motion accuracy, we adopt ParticleSFM(Zhao et al., 2022) to estimate camera trajectories from generated videos, with the use of Absolute Trajectory Error (ATE) to measure their differences from the ground truth. Relative Pose Error (RPE) is calculated to assess how well the relative motions between consecutive frames match the expected ones, including their translation (RPE-T) and rotation component (RPE-R). Specifically, we randomly sample 500 prompt-trajectory pairs from RealEstate10k (Zhou et al., 2018), and use them as references for calculating FVD and FID. Since SVD is an image-to-video model, we generate the first frames using Stable Diffusion (Rombach et al., 2022) based on text prompts. We also provide the results produced by vanilla SVD as a reference. For camera motion accuracy, we provide the evaluations on ground truth videos as a lower bound of these metrics.

Table 2: **Computational analysis of inference process,** evaluated under unified settings.

|  |  | SVD | MotionCtrl | CameraCtrl | CamTrol ($t_0 = 15$) |
|---|---|---|---|---|---|
| Max GPU memory(MB) |  | 11542 | 31702 | 26208 | 11542 |
| Time (s) | pre-process | - | - | - | 56 |
|  | inference | 11 | 32 | 42 | 8 |

(a) general scenes  (b) dynamics

Figure 4: **Generalization comparisons.** CamTrol can avoid domain collapse that arises from over-fitting on certain datasets and adapt to more general scenes (*Left*), meanwhile preserving video's dynamics while adhering to desired camera movements (*Right*).

## 4.2 COMPARISONS WITH STATE-OF-THE-ART METHODS

**Quantitative Evaluation**  Quantitative evaluations are shown in Table 1. In the table, the best results are in bold, and the second best are underlined. The performance of vanilla SVD (without motion control) is denoted as *SVD*, while the lower bound for motion metrics, provided by ground truth videos, is denoted as *Reference*. In terms of video quality, CamTrol attains performance comparable to methods finetuned on the RealEstate10k dataset. For motion accuracy, CamTrol also achieves the lowest score in ATE and RPE-T/R. The quantitative evaluations demonstrate CamTrol's ability to generate videos with both accurate camera motion and high visual quality.

**Qualitative Analysis**  Qualitative comparisons are illustrated in Fig. 3. The reference trajectory includes zoom, pan, and roll. While MotionCtrl and CameraCtrl fail to capture subtle camera motions, resulting in simple pan movements, CamTrol is able to follow the complex trajectory and generate videos with correct motion. We also evaluate the generalizability of different methods in generating more general scenes and dynamic content. The results are shown in Fig. 4. Since both MotionCtrl and CameraCtrl are finetuned on limited scenes (i.e., real estate videos) with static content, they have difficulty generalizing to other scenes, such as videos in non-realistic styles or videos that include people. As illustrated in Fig. 4, their motion controls in both situations are misaligned with the intended movements. Furthermore, finetuning on such datasets leads to a loss of dynamics, which is a crucial element in video generation. In comparison, CamTrol preserves most of the prior knowledge from video base models, enabling it to handle general scenes as well as generate dynamic content. Relevant videos are available in the supplementary materials.

**Computational Analysis**  We provide the computational analysis in Table 2, including the maximum GPU memory required and the processing time for all methods during inference. Evaluations are conducted under unified settings. We test at a resolution of $576 \times 320$ with 25 generation steps. The number of frames and the decoding size of SVD are set to 14. As a training-free method,

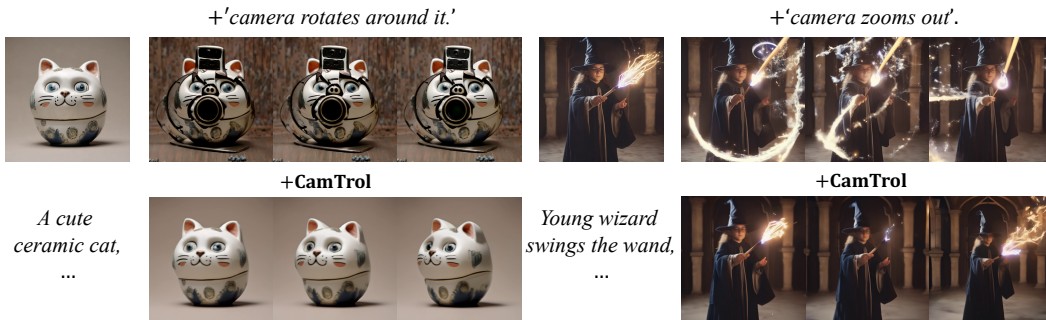

Figure 5: **Comparison with base model.** Controlling camera motion via prompt engineering rarely succeeds. Instead, CamTrol offers robust control to video's camera movements.

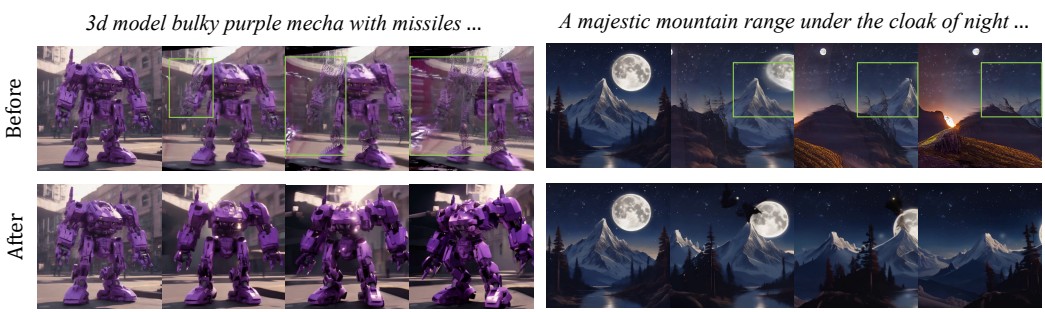

Figure 6: **Effectiveness of layout prior.** Layout prior guidance compensates for the voids (*Left*) and static elements (*Right*) in point cloud.

CamTrol requires no extra GPU memory during the inference process compared to the base model. This saves 10-20GB of GPU memory compared to other methods under the same circumstances, allowing it to run on a single RTX 3090. The major consumption of CamTrol comes from rendering multi-view images. Since this process is temporally sequential, i.e., $t_0 \rightarrow t_1 \rightarrow t_2$, a more parallelized approach may improve time efficiency. The results for $576 \times 1024$ resolution and more detailed settings can be found in the Appendix.

### 4.3 ABLATION STUDY

**Comparison to Base Model** To demonstrate that the changes in camera motion are attributed to our method rather than the innate capability of the video model, we conduct an ablation study to evaluate its effectiveness. We add prompts describing specific camera movements (e.g. *zooms out*) and let the video model interpret them on its own. The results are presented in Fig. 5. It can be observed that, even when provided with prompts indicating how the camera should move, the base model fails to produce correct results. In contrast, CamTrol is able to implement the designated motion control without any instructions from text prompts. In Table 1, the comparisons with vanilla SVD also demonstrate CamTrol's effectiveness.

**Effectiveness of Layout Prior** We conduct an ablation study to validate the effectiveness of layout prior guidance, demonstrating its necessity in two aspects: the completeness of missing areas and the dynamics of the generated video. In Fig. 6, we showcase frames before and after applying noise prior guidance. As camera pose changes, there are regions left unfilled in the point cloud, leading to blank spaces in rendered images (left part); Additionally, due to the static nature of point cloud, the rendered images remain stationary (right part). Layout prior guidance compensates for these flaws, ultimately producing videos with inpainted gaps and natural dynamics.

Table 3: **Quantitative effect of $t_0$.**

| $t_0$ | Video Quality | | | | Motion Accuracy | | |
|---|---|---|---|---|---|---|---|
| | FVD ↓ | FID ↓ | IS ↑ | CLIP-SIM ↑ | ATE ↓ | RPE-T ↓ | RPE-R ↓ |
| $t_0 = 20$ | 1079.88 | 68.52 | 7.14 | 0.3100 | 4.17 | 1.09 | 0.012 |
| $t_0 = 15$ | 778.46 | 68.06 | 7.05 | 0.3110 | 4.17 | 1.07 | 0.010 |
| $t_0 = 10$ | 754.14 | 67.98 | 7.00 | 0.3107 | 4.13 | 1.02 | 0.008 |

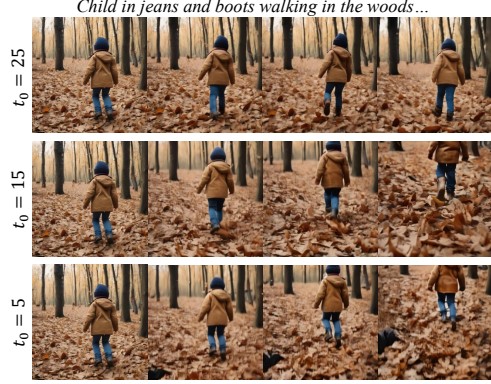

Figure 7: **Effect of $t_0$.** Larger $t_0$ encourages dynamics while smaller $t_0$ preserves camera movements (*Pedestal Down*).

Figure 8: **3D rotation videos at different motion scales.** CamTrol supports camera movements across multiple scales.

**Effect of Timestep $t_0$**   $t_0$ is a crucial factor that influences the trade-off between generated video's diversity and its faithfulness to camera motion requirements. To investigate its effect on the output, we conduct experiments with various $t_0$ values, the relevant results are shown in Fig. 7. As illustrated, videos generated with smaller $t_0$ tend to conform better to camera motion requirements, but suffer from decreased dynamics; On the other hand, larger $t_0$ leads to more plausible generations but fails to meet camera's requirements, since the latents at these timesteps carry more randomness. We also provide quantitative evaluations for different $t_0$ values in Table 3. As $t_0$ decreases, CamTrol produces videos resembling static, camera-moving scenes (which have lower FVD as we use RealEstate10k as the reference videos) with higher accuracy in motion control.

**Generalization to Diverse Situations**   Our proposed CamTrol can be seamlessly integrated into various scenarios, accommodating different base models, resolutions, and generation lengths, all in a training-free style. We present visual results of its applications under different settings, including CogVideoX (Yang et al., 2024b) (diffusion transformer model, $720 \times 480$ resolution, 49 frames) and VideoFusion (Luo et al., 2023) (decomposed diffusion process, $128 \times 128$ resolution, 16 frames), in Fig. 10. Our approach remains effective when applied to alternative video base models, resolutions, and generation lengths, demonstrating its strong robustness and generalizability.

## 4.4 FURTHER APPLICATIONS

**3D Rotation Videos**   One of the main advantages of our method is that it can generate videos with rotating movements and produce outputs similar to those of 3D generation models (Voleti et al., 2024; Melas-Kyriazi et al., 2024). While these 3D models require large-scale training on 3D datasets and can only handle inputs in specific styles, our approach is capable of dealing with any type of image and achieving this in a completely zero-shot manner. An example is shown in Fig. 8.

**Hybrid and Complex Camera Movements**   By combining basic camera trajectories, CamTrol can support hybrid camera movements and produce videos with cinematic appeal. In addition, explicit motion modeling equips CamTrol with the ability to handle trajectories of precise extrinsics, allowing it to generate videos with arbitrarily complicated camera movements. Relevant results can be found at **https://lifedecoder.github.io/CamTrol/**.

*A snow-capped mountain peak towers above a tranquil alpine lake, mirrored perfectly in its glassy surface.*

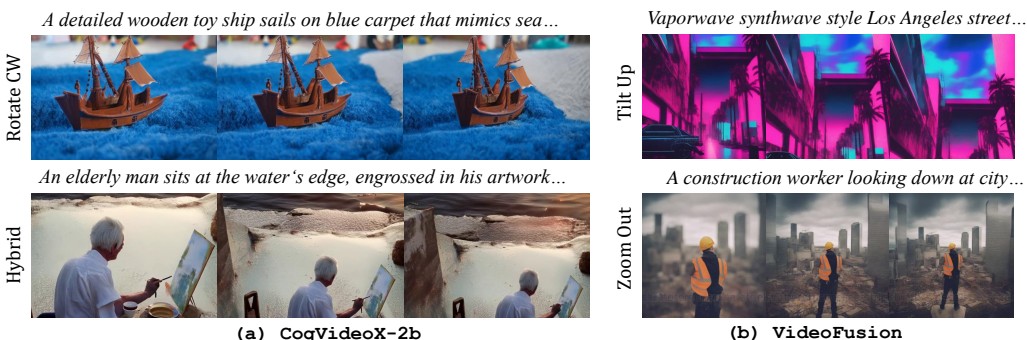

Figure 9: **Multi-trajectory video generation.** CamTrol is able to generate videos with different camera motions for the same scene.

Figure 10: **Applied to CogVideoX (Yang et al., 2024b) and VideoFusion (Luo et al., 2023).** CamTrol can be plug-and-play under various situations, accommodating different base models, different resolutions, different generation lengths, all in a training-free manner.

**Multi-trajectory Video Generation**  One natural application of our method is to generate multi-trajectory videos for a single scene. Since the point cloud remains fixed once the reconstruction has finished, the spatial consistency between different trajectories is guaranteed, making it easy to produce multiple camera-moving videos of the same scene. We demonstrate this application in Fig. 9. More results on multi-trajectory video generation can be found in the supplementary materials.

**Camera Motion at Different Scales**  CamTrol supports camera movements at controllable scales. By specifying different magnitudes of the camera's extrinsic matrix, the rendered images will exhibit varying degrees of motion, resulting in videos with distinct scales of camera movements. This further demonstrates the powerful controllability of CamTrol and provides a new pathway for customized camera control of video generation. We provide relevant visualization in Fig. 8.

## 5 CONCLUSION

In this paper, we propose a training-free and robust method **CamTrol** to offer camera control for off-the-shelf video diffusion models. It consists of a two-stage procedure, including explicit camera motion modeling in 3D point cloud space and video generation utilizing the layout prior of noisy latents. Compared to previous works, CamTrol does not require additional finetuning on camera-annotated datasets or self-supervised training via data augmentation. Comprehensive experiments demonstrate its superior performance in both generation quality and motion alignment against other state-of-the-art methods. Furthermore, we show the ability of CamTrol to generalize to various scenarios, as well as its impressive applications including unsupervised generation of 3D video, scalable motion control, and dealing with complicated camera trajectories.

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

## A    MORE RESULTS ON CAMERA CONTROL

We present additional qualitative results of CamTrol on our demo page:

**https://lifedecoder.github.io/CamTrol/**

This demo page includes CamTrol-generated videos including basic camera motions (including *Zoom, Tilt, Pan, Pedestal, Truck, Roll, Rotate, Hybrid, Complicated*, detailed definitions are in Appendix B), hybrid motions (*Zoom In first, then Pedestal Up*, *Zoom Out + Pedestal Up + Truck Left + Tilt Down + Pan Right*), and complicated motions generated from precise camera extrinsics (extracted from RealEstate10k (Zhou et al., 2018)).

In addition, it contains 3D rotation videos generated unsupervised from video base models (both objects and scenes). These outputs share similarities with those of 3D generation models, as they all exhibit in a turning-table-like way, with the camera rotating around objects. The difference here is that the 3D model, as trained on specific datasets, can only generate outputs in certain styles, *e.g.*, single static object with no background. Instead, our model can handle arbitrary images as input and generate rotating videos with appropriate dynamics. From this perspective, our method could be seen as an infinite source of 3D data. By employing our method with stronger backbones, video foundation models could truly become the largest source of 3D data, as they should be.

Furthermore, it contains additional results discussed in ablation studies, e.g., controlling camera motions at different scales, and the effectiveness of using layout prior compared with the raw output given by the video base model.

## B    DEFINITIONS OF BASIC CAMERA MOTIONS

We refer to the terminology in cinematography to describe different camera motions, the definitions of each type are detailed in Table 5.

To obtain consistent images from multiple views, we set camera motion as a trajectory of extrinsic matrices $\{\mathbf{P}_1, ..., \mathbf{P}_{N-1}\}$, each consisting of a rotation matrix and a translation matrix, representing the camera pose and position. For hybrid motions, CamTrol supports both spatial (i.e., performing several basic motions simultaneously) and temporal (i.e., concatenating basic motions sequentially) combinations. In the case of complicated trajectories, one can directly use precise parameters for camera's extrinsic matrix as input to control video's motion. Additionally, if these parameters are not available, a reference video with corresponding move can also serve as input. With the help of SFM algorithms (e.g., COLMAP [1], ParticleSFM (Zhao et al., 2022)), camera motion can be easily estimated and used for imitation. In this sense, CamTrol can also be seen as an unsupervised method to achieve video motion transfer.

## C    VISUAL COMPARISONS WITH STATE-OF-THE-ARTS

Sec. 3 showcases some qualitative comparisons between CamTrol and different state-of-the-art approaches. For better visualization and comparison, we provide more video results in the supplementary materials.

## D    IMPLEMENTATION DETAILS

For text prompt input, we use Stable Diffusion v2-1 [2] or Stable Diffusion XL [3] to generate the initial image. The inpainting model we apply is Stable Diffusion inpainting model proposed by Runway [4], and the backward step of inpainting is set to 25. We use ZeoDepth [5] as depth estimation model. The classifier-free guidance scale follows the default setting of the base models themselves. We use

---

[1]https://github.com/colmap/colmap

[2]https://huggingface.co/stabilityai/stable-diffusion-2-1

[3]https://huggingface.co/stabilityai/stable-diffusion-xl-base-1.0

[4]https://huggingface.co/runwayml/stable-diffusion-inpainting

[5]https://github.com/isl-org/ZoeDepth

Table 4: **Computational analysis of on $576 \times 1024$.**

|  |  | SVD | MotionCtrl | CameraCtrl | CamTrol ($t_0 = 15$) |
|---|---|---|---|---|---|
| Max GPU memory(MB) |  | 34236 | 71096 | - | 34236 |
| Time (s) | pre-process | - | - | - | 149 |
|  | inference | 32 | 54 | - | 22 |

Table 5: **Definitions of camera motions.** We adopt the terminology from cinematography to describe different camera movements.

| Camera Motion | Directions | Definition |
|---|---|---|
| Zoom | In
Out | Camera moves towards or away from a subject. |
| Tilt | Up
Down | Rotating the camera vertically from a fixed position. |
| Pan | Left
Right | Rotating the camera horizontally from a fixed position. |
| Pedestal | Up
Down | Moving the camera vertically in its entirety. |
| Truck | Left
Right | Moving the camera horizontally in its entirety. |
| Roll | Clockwise
Anticlockwise | Rotating the camera in its entirety in a horizontal manner. |
| Rotate | Clockwise
Anticlockwise | Moving the camera around a subject. |
| Hybrid | Combinations | Spatial and temporal combinations of basic motions. |
| Complicated | Arbitrary | Camera extrinsic matrices or reference videos as input. |

SVD's default setting of 6 fps for video generation, and process reference videos to 6 fps for FVD and FID calculations. The complete procedure of CamTrol is elucidated in Algorithm 1.

For computational analysis, we set both the number of frames and the decoding size to 14, and the generation steps to 25. We do not apply xformers in any of the approaches. The computational analysis at $576 \times 1024$ resolution is shown in Table 4. SVD-based CameraCtrl only supports a resolution at $320 \times 576$.

## E    CHOICE OF 3D REPRESENTATION

In Sec. 3.1, we designate point cloud as the intermediate 3D representation for explicit camera motion modeling. Doubts may arise as to why we do not use a more complex 3D representation which might be more useful. Here we take the most prevalent 3D representation–3D Gaussian Splatting [6], as an example to elaborate on the benefits of using point cloud in three aspects:

Firstly, concerning the input, point cloud is able to construct the entire scene from one single input image combining techniques of depth estimation and inpainting. GS, though also an explicit 3D representation, requires optimization based on images from different views, which means it is neither capable of handling single input image, nor can it leverage 2D inpainting to gradually complete the scene.

---

[6]https://repo-sam.inria.fr/fungraph/3d-gaussian-splatting/

---

**Algorithm 1:** Training-free camera control for video generation

---

**Input:** Text prompt $p$, camera motion $\mathbf{P}$, input image $\mathbf{I}_0$ (*optional*).

```
// Stage I: Camera Motion Modeling:
```

1  **for** *i=1,...,N-1* **do**
2  $\quad$ $\tilde{\mathbf{I}}_i = \text{inpainting}(\mathbf{I}_{i-1}, \mathbf{P}_i, p)$ ;
3  $\quad$ $\tilde{\mathbf{D}}_i = \text{depth}(\tilde{\mathbf{I}}_i)$ ;
4  $\quad$ **while** *not converged* **do**
5  $\quad\quad$ $d_i = \text{argmin}_d \left( \sum_M \left\| \phi([\tilde{\mathbf{I}}_i, d\tilde{\mathbf{D}}_i], \mathbf{K}, \mathbf{P}_i) - \mathcal{P}_{i-1} \right\| \right)$
6  $\quad$ **end**
7  $\quad$ $\mathcal{P}_i = \phi([\tilde{\mathbf{I}}_i, d_i\tilde{\mathbf{D}}_i], \mathbf{K}, \mathbf{P}_i)$ ;
8  $\quad$ $\mathbf{I}_i = \psi(\mathcal{P}_i, \mathbf{K}, \mathbf{P}_i)$ ;
9  **end**

```
// Stage II: Layout Prior Generation:
```

10  $\mathbf{V}_0 \leftarrow \{\mathbf{I}_i\}_{i=0}^{N-1}$ ;
11  Sample random noise $\epsilon \sim \mathcal{N}(\mathbf{0}, \mathbf{I})$ ;
12  Motion inversion $\mathbf{V}_{t_0} \leftarrow \sqrt{\bar{\alpha}_{t_0}}\mathbf{V}_0 + \sqrt{1 - \bar{\alpha}_{t_0}}\epsilon$ ;
13  **for** *t=$t_0$,...,1* **do**
14  $\quad$ $\mathbf{V}_{t-1} \leftarrow \sqrt{\alpha_{t-1}} \left( \frac{\mathbf{V}_t - \sqrt{1-\alpha_t}\epsilon_\theta^{(t)}(\mathbf{V}_t)}{\sqrt{\alpha_t}} \right) + \sqrt{1 - \alpha_{t-1} - \sigma_t^2}\epsilon_\theta^{(t)}(\mathbf{V}_t) + \sigma_t\epsilon$
15  **end**

---

Secondly, from the aspect of time, point cloud can directly lift 2D points onto 3D spaces, while 3DGS demands optimization for each scenario. As a training-free approach, our method takes almost no time to generate a video after multi-view images are acquired, but would need more time if 3DGS were applied.

Lastly, from the task itself, the goal of stage I is not to precisely reconstruct the 3D scene but rather to offer a rough layout guidance. In this context, rendered images from point cloud are sufficiently qualified, and no further refinement of the 3D reconstruction is necessary.

From the analysis, point cloud is quite qualified serving as a rough layout guidance in a relatively quick speed, without any further optimization or redundant multi-view images as input. The use of point cloud allows our algorithm to be completely training-free, while still being able to produce camera-moving videos quickly with only one single image or text prompt as input.

## F   DETAILS ABOUT GENERATING 3D ROTATION VIDEOS

3D generation models (Voleti et al., 2024; Melas-Kyriazi et al., 2024; Shi et al., 2023; Chen et al., 2024; Han et al., 2024) are trained on highly-regulated 3D datasets, these datasets are hard to collect, and include only a limited variety of data types (e.g., single static objects with no background). As a consequence, 3D generation models exhibit a very constrained output distribution and can only produce results in certain styles. CamTrol avoids these problems of arduous data collection, laborious finetuning, and output collapse by utilizing the layout prior hidden in video foundation models. Not only does it require no training, but this advantage also benefits CamTrol from inheriting most of the prior knowledge inside video foundation models, such as the diversity of scenarios, the dynamics of moving objects, temporal consistency, etc. Thus, CamTrol is able to generate dynamic 3D content, both objects and scenes, in a totally unsupervised and training-free manner, this is what other methods cannot achieve yet.

Compared with regulated datasets, the problem of processing wild pictures in 3D is that some of the parameters are unknown. In Sec. 3.1, we mentioned that the camera intrinsic matrix $\mathbf{K}$ and the initial extrinsic matrix $\mathbf{P}_0$ are set by convention, as they are usually intractable. Another crucial parameter concerning 3D video generation is the distance between the camera and the content of

input image (denoted as $f$), note that input image could be synthetic or real. Considering that most camera rotations are performed around the center point, we extract a patch from the very center of the input image, conduct depth estimations on it, and define the distance $f$ as the average depth. The rotation and translation matrices of camera movement can be formed as:

$$\mathcal{R}_y = \begin{bmatrix} \cos\theta_i & 0 & -\sin\theta_i \\ 0 & 1 & 0 \\ \sin\theta_i & 0 & \cos\theta_i \end{bmatrix}, \quad t = \begin{bmatrix} f\sin\theta_i \\ 0 \\ f - f\cos\theta_i \end{bmatrix},$$

$$\text{where} \quad f = \frac{1}{|P|} \sum_{(j,k)\in P} D(x_0 + j, y_0 + k). \tag{6}$$

Here, $i \in [0, N-1]$ and $\theta_i$ refer to the rotation angle around the $y$ axis at step $i$, $D$ denotes the depth estimation of the image, and $P$ represents the patch around the central point $(x_0, y_0)$. In our experiment, we choose $(j, k) \in [-10, 10]$ as the size of the patch.

## G    MOTION BLUR, PROBLEMS AND SOLUTIONS

Videos produced by CamTrol need to satisfy certain camera movements, and some drastic perspective changes might cause visible trails, recognized as motion blur of objects or scenes. This phenomenon will appear to be more indispensable when the video base model has a relatively small generation length (e.g., 16 frames) as well as the motion scale becomes larger (e.g., tilt up by 60 degrees or more). To avoid blur issues when controlling video camera motion, we propose several solutions as follows:

1. According to the analysis above, the blur issue is caused by limited generation frames and large camera movements, thus the most intuitive solution is to either reduce motion scale or use a more capable generation model. For severe perspective changes, the optimal approach would involve employing video foundation models that support larger generation lengths (e.g., CogVideoX (Yang et al., 2024b) supports generating videos with 49 frames). This allows the model to manage motions of equivalent magnitude while experiencing a smaller moving range between adjacent frames, thereby bringing effective alleviation to the blur problem.

2. One can also stack the results of multiple generations to form a complete output, i.e., treating the last frame of the previous generation as the starting frame for the next and integrating them as a whole. This approach is more suitable when using an image-to-video (I2V) base model. Since most open-source video foundation models are text-to-video (T2V), one may consider increasing the step of camera motion inversion $t_0$, which guarantees more fidelity towards input images' content (and motion).

3. Besides the above two approaches, applying frame interpolation to the output is also a common and convenient choice. Many off-the-shelf frame interpolation models are open-source and can be found on GitHub.

4. Lastly, if the video length cannot be altered, it may be necessary to increase the fps of the generated videos for better visual quality. Although a larger frame rate leads to a shorter video duration, it simultaneously makes the visual persistence brought by motions less pronounced, which reduces blur visually.

In our experiments, we take the raw output in all settings.

