# OpenReview forum: "Training-free Camera Control for Video Generation"
_ICLR.cc/2025/Conference — ICLR 2025 Poster_

### Official Review · Reviewer_14Er · 2024-10-21

**Soundness:** 3
**Presentation:** 2
**Contribution:** 2
**Rating:** 5
**Confidence:** 5

**Summary:**

This paper proposes CamTrol, a training-free method for camera control in pre-trained video diffusion models. While previous works mainly train a camera control module using a fine-tuning dataset with camera annotations, this work does not require any external dataset to control the camera. The pipeline uses 3D point clouds and inversion to guide the generation process, enabling flexible camera control. CamTrol outperforms previous works which rely on external data, while enabling metric scale control based on the metric depth estimator.

**Strengths:**

- Control over motion strength: The method can synthesize videos with different camera motion strengths. Previous methods struggle due to having no knowledge of scale. Because the method uses a metric depth estimator, this problem should be fixed.

**Weaknesses:**

- Scene inconsistency: The scene changes when the camera moves. When looking at the supplementary website, “Tilt Down” changes the top part of the scene or “Pedestal Down” changes the whole arrangement of the scene. But the point of camera control is that the scene remains intact, so in the best case the scene could be reconstructed from the video.
- Inaccurate camera: When looking at the supplementary website, the cameras are not that accurate. E.g., “Pan Left” also moves forward in the scene and doesn’t look like panning but rotating on the spot.
- Missing computational costs analysis: The approach seems very expensive. The authors motivate that previous works fine-tune some camera control modules on a camera annotated dataset which requires additional training. However, those approaches can do inference with basically zero computational overhead after training. This optimization-based approach requires computational overhead for each inference run and it is not clear how significant that is. E.g., if a user has to wait tens of minutes to hours to generate one result, there is a trade-off reached after a few prompts where it is more practical to train a camera control module once and then use it without overhead. It’s important to analyze both optimization time and memory usage.
- Missing quantitative ablations: The ablations mentioned in Sec. 4.3 are missing quantitative comparisons. The depth optimization seems to change nearly nothing in Tab. 2. When looking at relative differences between metrics, the metrics differences are basically 0.
- Writing quality: While the paper starts ok in terms of grammar, it gets worse throughout the paper and it makes the paper hard to read, especially the method section. I recommend putting the paper through some grammar correction tool.
- Unclear motivation of generalizability: The paper mentions in the introduction that methods fine-tuned on RealEstate10K suffer from generalizability issues. But there are no experiments or visuals which demonstrate this. Does the method really work better when cameras totally different from the training distribution are used?

**Questions:**

I currently rate this paper below the acceptance threshold. I have some concerns about the result quality and moreover it is not clear how large the computational overhead at inference time is compared to just having some simple embedding module as in MotionCtrl and CameraCtrl.
I would like authors to address following questions:
- Why does the scene often change with camera movement?
- Why is the camera often not that accurate even for simple movements?
- What is the computational overhead at inference time?

I am open to adjusting my rating based on the rebuttal.

---

> ### Author Response · Authors · 2024-11-23
> **Rebuttal of CamTrol to Reviewer 14Er**
>
> Thank you for your meticulous review and insightful comments of our work.
>
> We start with some explainations about the weakness:
>
> **W1**: Camera motion should keep the scene intact and should not change the arrangement.
>
> **A1**: Yes, scenes should be kept intact when camera moves. Simultaneously, the other goal we want to achieve is to have more dynamics in the generated videos, just like what we record through our phones in the real world. And along with the camera movements, the arrangment of scenes can be changed. In this setting, the whole scene can be reconstructed from video by some 4D reconstruction algorithms.
>
> **W2**: Inaccurate camera motion.
>
> **A2**: We follow the terminology in cinematography to describe different camera movements, where the definition of "pan" refers to "rotating the camera horizontally from a fixed position", and the motion that moves horizontally is described as "truck". The definitions of camera motion are presented in Line 178 and Line 765 of the paper.
>
> **W3**: Missing computational analysis. The approach seems very expensive.
>
> **A3**: We add the computational analysis into the revised version of our paper in Table. 2 and Line 375-412 (the updated parts are highlighed in blue). We test the GPU memory and time requires in inference process for different methods, the results are shown as below:
>
> **Table2. Computational Analysis**
> ||SVD|MotionCtrl+SVD|CameraCtrl+SVD|CamTrol+SVD|
> |-|-|-|-|-|
> |GPU memory (max)/MB|11542|31702|26208|11542|
> |Time (s)  - preprocess |-|-|-|56|
> |Time (s) -  inference|11|32|42|8|
>
> The evaluations are conducted under unified settings for all approaches. As a training-free method, our method requires no extra GPU memory during inference process compared with base model. This makes it save 10-20GB memory than other finetuned methods under the same circumstances, and makes it applicable running on a single RTX 3090 GPU. The major consumption of our method comes from the time rendering multi-view images. Even so, the time is acceptable as the other two methods also need considerable time.
>
> **W4**: Missing quantitative ablations.
>
> **A4**: We add two more quantitative ablations in the revised version of paper: 1) Ablations to vanilla SVD. We evaluate the quantitative performance of vanilla SVD to demonstrate the effectiveness of our approach in camera motion control. 2) Ablations on t_0. We conduct quantitative experiments with different t_0 to show how it affect output videos in motion accuracy.
>
> **W5**: Writing quality in methodology part.
>
> **A5**: Sorry for the confusion. When writing the paper, we also felt some parts are difficult to describe, especially the illustration about latents' casuality and error-resilience towards generated outputs. We will use some grammar tools to refine them.
>
> **W6**: Unclear motivation of generalizability.
>
> **A6**: The generalizability refers to camera control ability of videos from other domains. We add comparison results of generalizability in Fig. 4 and Line 366 - 373 in the revised version of paper.  Since both MotionCtrl and CameraCtrl are finetuned on limited scenes (i.e., real estate videos) with static content, they can hardly generalize to other scenes, such as videos in non-realistic style or videos that include people. They fail to offer accurate camera control, nor can they generate content with dynamic movements. In comparison, CamTrol preserves most of the prior knowledge from video base models, enabling it to handle general scenes as well as generate dynamic content. We put these results in the updated supplementary materials, in the end of the qualitative comparison file named "rebuttal_1_qualitative_comparisons_new".
>
>
> Secondly, we will answer the questions as follows:
>
> **Q1**: Why does the scene often change with camera movement? Why is the camera often not that accurate even for simple movements?
>
> **A1**: The reason for this two issues is because we use smaller t_0. If we set t_0 to a larger value, the camera motion will be more accurate and the scene will be more consistent, but it will lead to loss of video dynamics. Setting smaller t_0 will allow the generated videos with more dynamics, while sacrificing some accuracy in motion control. However, these motions remain close to the original trajectory without significant deviation.
>
> Methods that are finetuned on camera-annotated datasets have higher accuracy in motion control, but they cannot handle scenes outside the training domain (like MotionCtrl and CameraCtrl can only handle scenes of real estates), neither can they deal with dynamic contents. When users input a camera motion, such as "Zoom In" into our method, we assume they prefer a well-generated, dynamic, realistic video, even the generated motion has little diviation from given one, it still looks natural. Instead, a video that follows exactly the input motion, but with limited scenes or static objects may not be preferred.
>
> **Q2**: This question refers to the same issue with W3.

---

> > ### Comment · Reviewer_14Er · 2024-11-26
> >
> > Thanks for the response! I have some remaining concerns about the scene inconsistency. The authors basically say that scene consistency is compromised for having some scene motion maintained with accurate camera control.
> > Also I do not buy the argument that MotionCtrl and CameraCtrl can only show in-domain videos (RE10K) while this approach can do out-of-domain dynamic ones. Most results and comparisons shown are from static scenes. Also the new CVD results in the supplementary are all static scenes.
> >
> > I am not sure if this approach can handle dynamic video generation with camera control well. Even the CogVideoX motion seems to be degraded a lot compared to what the base model can do. Is it possible to show more dynamic results? I understand that MotionCtrl and CameraCtrl have the issue of showing very static scenes, but the problem does not seem to be solved here.

---

> ### Author Response · Authors · 2024-11-26
> **Second rebuttal to Reviewer 14Er**
>
> We thank reviewer 14Er for checking the results in the supplementary materials and for the response.
>
> Please allow us to explain further about the concerns:
>
> 1. The issue of scene consistency, and also the problem of motion degradation compared with base models.
>
> For most video foundation models, their training datasets (e.g., Webvid, UCF-101) don't contain videos with such quick, severe camera movements like what our method produces. Referring to our anonymous demo page, we'll find that many videos have completely different scenes in merely 16 frames with camera moving. Some of the results in 3D part even have movements as drastic as SV3D can produce, and SV3D is a 21-frame model finetuned on 3D data. We assume this is the reason that degradations can occur, as we want to let base models produce quick, severe camera moving videos, where they originally only produce videos with moderate camera motion. This problem could be solved if we had a base model that can originally produce drastic moves, or had a annotated dataset that contains generic, diverse scenes to finetune base model. Yet we have neither of these two in our community.
>
> 2. The dynamic control of finetuned methods using limited datasets (MotionCtrl and CameraCtrl) and of CamTrol.
>
> We consider the domain collapse issue of MotionCtrl and CameraCtrl not come from the operation of "finetune". If we have the proper datasets, finetuning generally can give the best performance compared with other heuristic methods. The problem is that current camera-annotated datasets only contain very limited scenes, i.e., real estate videos with static content. As generative models are basically mapping functions from one distribution to another, the original output distribution of video base model will shrink to a smaller, contrained distrbution after finetuning on these datasets. For our approach, it does not change the output distribution of video base models (as it only uses the inference process of them), but only gives them a guidance of how output videos should be formed (i.e., in certain layouts). This allows it to perserve more prior knowledge inherited from foundation models (such as the generalization of scenes and dynamics of generated content) compared with methods finetuned on mentioned datasets.
>
> We had some examples in Fig.4 of the revised paper to show the performance of MotionCtrl and CameraCtrl dealing with generic and dynamic scenes. We also updated some extra comparisons in the supplementary materials, in a new html file named "second_rebuttal_dynamic_comparisons.html". We need to clarify that the open-sourced SVD model itself does not perform very well in generating dynamic content (it exhibits mostly static videos, which could be referred to through: https://huggingface.co/stabilityai/stable-video-diffusion-img2vid) and has limited capacity generating videos with drastic camera move, thus all four methods based on SVD (including SVD itself) struggle to produce consistent, perfect results, but our method performs better than others in generating dynamic content and generalizing to more common scenes. Applying to a more powerful, dynamic base model, it will show more dynamics in generated videos. We provide more dynamic videos produced by CamTrol in the newly added file named "second_rebuttal_more_dynamics". There are also high-dynamic results presented in our anonymous demo web page. These files contain many camera-moving videos with quite high dynamics in their generated content. We believe that no approach, to our knowledge, can generate such high-dynamic video for generic scenes (e.g., include human, animals, natural scenes, even 3D objects from 3D datasets) with camera control.

---

> > ### Comment · Reviewer_14Er · 2024-12-02
> >
> > Some concerns are addressed, except the quality of the results. The results even for CogVideoX are rather static. Also there are not a lot of results. I highly recommend creating a grid of results where the camera is shared across many prompts to show that one camera works across prompts for different dynamics. And also more difficult cameras would be great to show, panning and zooming are very simple and don't even need 3D. So more complex combinations of rotations and translations would be helpful. I am not sure if all of this is not too much for a rebuttal to be convincing, since November 27th it's mainly a discussion period anyway and not to provide new results, as the supplementary can not be updated. But I highly recommend to include more visual results, comparisons, dynamic scenes, complex cameras, and cameras shared across prompts. Currently the results are not fully convincing to see the benefit over previous works. I will increase to 5, but do not feel comfortable going higher.

---

> ### Author Response · Authors · 2024-12-02
> **Rely to Reviewer 14Er**
>
> We thank reviewer 14Er for raising the score.
>
> We arranged the results the same way that reviewer 14Er recommended in our anonymous demo page, i.e., camera is shared across many prompts to show that one camera works across prompts for different dynamics, especially in the "hybrid and complex trajectories" and "3D rotation-like generation" parts, and these results are all complex combinations of rotations and translations. There are 40+ videos presented in these two parts on the anonymous demo page, all with complex (combinations of rotations and translations, which reviewer 14Er recommended), dynamic moves, arranged in the mentioned way (one camera works across prompts for different dynamics).
>
> For the benefit over previous works, we recommend reviewer 14Er to check the demos produced by these methods, MotionCtrl: https://wzhouxiff.github.io/projects/MotionCtrl/ and CameraCtrl: https://hehao13.github.io/projects-CameraCtrl/, to judge whether our method produce more dynamic results than them. Most videos produced by MotionCtrl and CameraCtrl are very static, as can be seen from the exhibited results on their websites. If one claims that demos on webpage do not fully show the performance of methods, we also include comparions under the same circumstances in the supplementary material, to demonstrate their weakness in handling generic, dynamic scenes. We strongly recommend reviewer 14Er to check previous methods' websites for a judgement that whether our method has benefit over them, in both training cost and generating performance.
>
> Best regards,
>
> Authors

---

> > ### Comment · Reviewer_14Er · 2024-12-02
> >
> > Thanks for the reply. However, I do not find the same scenes used for different cameras. You use one camera for multiple scenes, but at the same time, change the scene when using a new camera. The best way is to show a grid of prompts/images, generate the same camera in this grid and show multiple grids with different cameras. So it is clear that the scenes all remain consistent, while only the scene motion would not be synchronized.
> > I agree that MotionCtrl and CameraCtrl are very static due to training on RealEstate10K but the scenes shown are also rather static. Hence, I do not see a big improvement in the visual results. This is the main reason, why I am not super convinced by the results. The results look somewhat similar to previous works from one year/half a year ago. And I am expecting a bigger step in the motion quality problem to justify this as a contribution. As otherwise the task has been investigated already. It's not clear what existing issue has been significantly improved compared to MotionCtrl and CameraCtrl to justify acceptance.

---

> ### Author Response · Authors · 2024-12-02
> **Reply to Reviewer 14Er**
>
> We thank reviewer 14Er for the response.
>
>  Reviewer 14Er might expect a bigger step in the motion quality problem to justify this as a contribution, however, particularly dynamic video generation is not what our method aims for. The purpose of our method is to provide a training-free (don't need finetuning to run) , low-cost (low in GPU memory) tool that can be directly adept to video diffusion models (it can work on most video diffusion models) to control video's camera motion. The generalizability and dynamics, in which aspects our method has better performance than others, are not initial purpose our method is proposed, but side-advantages of how our method utilizes the prior knowledge in video models. We also disagree with the saying that this task has been investigated already, methods like MotionCtrl and CameraCtrl have limited capacity which reviewer 14Er also agreed, what they mainly did was adding some layers to inject camera trajectory, then used camera-annotated datasets to finetune video model. Our method is the first one to achieve camera control on video diffusion models that does not require any finetuning and rely on video model's extensive knowlegde rather than finetuning for generating camera-moving videos.
>
> Reviewer 14Er also said it's not clear what existing issue has been significantly improved compared to MotionCtrl and CameraCtrl. We recommend reviewer 14Er to think about what our method can achieve while MotionCtrl and CameraCtrl can't, for which we can claim many: training-free and do not require finetuning, directly work on almost any video diffusion model, have more generalizability in generic scenes, low-gpu costs... . We also have some 3D generating results on the anonymous demo page, in the "3d rotation-like generation", where we generate camera-moving videos on 3D data from Omniobject3D, unsupervisedly. We believe previous methods like MotionCtrl and CameraCtrl cannot produce these kinds of results due to their limited capacity, and even 3D generative models like SV3D need finetuning on 3D dataset to produce these results. These are all benefits our method has over previous methods.

---

> ### Author Response · Authors · 2024-12-02
> **Another reply to Reviewer 14Er**
>
> Dear Reviewer 14Er,
>
> We have some new results updated in this comment, which may be helpful to the argument of visual quality:
>
> > I do not see a big improvement in the visual results.
>
> About the visual quality, we did conduct a user study among different methods, we did not put it in the paper. We feel necessary to put it here:
>
> ||Visual Quality|Motion Accuracy|
> |-|-|-|
> |AnimateDiff|0.82|1.12|
> |MotionCtrl|2.03|2.00|
> |CameraCtrl|1.08| 1.01|
> |CamTrol|2.06| 1.86|
>
> (Details of this user study: We collect 7 text prompt for 8 basic camera trajectories (zoom in/out, pedestal up/down, truck left/right, roll clockwise/anticlockwise), yield 56 samples for each method to generate. Users are asked to sort the four videos displayed randomly generated by four methods, and rate them from the highest score 3 to the lowest 1 (the higher the better). We calculated the average score attained by each method, forming the table up above. We have 17 people took part in the user study.)
>
> The results of user study prove that the visual quality of our method is acknowlegded by most users. If reviewer 14Er still feel not convinced by the results given by user study, there are also other reviewers who have positively recognized the generated quality of our work: Reviewer syyx stated that "the experiments and demo show good results", reviewer 4R7b commented that our method "outperforms competing methods in both perceptual quality and motion alignment, particularly in complex camera movements", and reviewer WcRk said our results "addresses the concerns of dynamics".
>
> >I agree that MotionCtrl and CameraCtrl are very static due to training on RealEstate10K but the scenes shown are also rather static.
>
> We kindly ask reviewer 14Er to provide some concrete examples for this, i.e., the results of our method that are rather static compared with MotionCtrl and CameraCtrl.
>
> > The results look somewhat similar to previous works from one year/half a year ago.
>
> On their websites, we did not see MotionCtrl and CameraCtrl produce results with
>
> 1) such diverse scenes with dynamic content, including human with drastic moves **(e.g., playing drums, walking, playing guitar)**, dynamic generic scenes **(e.g., street with racing cars, burning flames)**, moving animals **(e.g. walking, jumping)**. We haven't seen previous works like MotionCtrl and CameraCtrl generating these kinds of results.
>
> 2) rotation videos for 3D data, which are exhibited in the "3d rotation-like generation" part of our anonymous webpage, **here we generate 3D rotation videos unsupervisedly using a single rendering image of object from OmniObject3D.** In fact, we believe no method before has achieved unsupervised, training-free 3D generation merely from video models.
>
> We hope these updated results as well as the upper claims could help address the issue of visual quality as well as the novelty compared with previous works, and we look forward to further reply from reviewer 14Er.
>
> Best regards,
>
> Authors

---

> > ### Comment · Reviewer_14Er · 2024-12-02
> >
> > Thanks for the response. I wonder why there is not the same scene shown with multiple trajectories as I outlined above? To show that the camera control actually works. That concern was somehow ignored.

---

> > > ### Author Response · Authors · 2024-12-03
> > > **Reply to Reviewer 14Er**
> > >
> > > We thank reviewer 14Er for the reply.
> > >
> > > The reason that we didn't talk about it is 1) We assume the main concern of reviewer 14Er is the visual quality, which reviewer 14Er also have claimed in the previous comment; 2) This is a recommendation of how to arrange exhibited results from reviewer 14Er, as the deadline for submitting supplementary materials has passed, we couldn't make any changes to the demo page. Thus we mainly focused on the explanation of visual quality. We will explain the layout problem here:
> > >
> > > > The best way is to show a grid of prompts/images, generate the same camera in this grid and show multiple grids with different cameras.
> > >
> > > Yes, it is a clear way to exhibit the results. Currently we have lines of videos with same camera trajectories in our demo page. When constructing the demo page, we thought to exhibit as much diverse scenes to the viewers, and thus we made effort selecting prompts of different styles and generating videos of different styles (we have videos in mythological style, pixel art, ultrarealistic, photorealistic... ), to demonstrate our method can handle videos with diverse scenes. It was in the rebuttal that we provide videos which have the same scenes but with different camera moves (CVD-like results). We haven't integrate them into our demo page as the deadline of submitting them has passed, but the results we updated in the supplementary (file "rebuttal_2_CVD-like_results") demonstrated our method is capable of generating videos like this.
> > >
> > > Those CVD-like videos are another important demonstration of our method's capabilities. Like the generalizability and dynamic of scenes, it is a side-advantage brought by how our method utilizes the prior knowlegde of video models, but could roughly handle problems that other methods spend most of their papers addressing (e.g., CVD). We will put these results on our demo page, following the layout advice of reviewer 14Er to show clearly the consistency of the same scene while camera moving.

---

> > > ### Author Response · Authors · 2024-12-03
> > > **Another reply to Reviewer 14Er**
> > >
> > > Dear reviewer 14Er,
> > >
> > > In previous comments, reviewer 14Er commented that "The results even for CogVideoX (+CamTrol) are rather static". We feel necessary to have some raw outputs given by CogVideoX to clarify that our method is not the cause of static. We add some other results of dynamic comparisons, including the raw videos generated by CogVideoX-5b, and the videos generated by CamTrol+CogVideoX-5b.
> > >
> > > As we could not upload any supplementary materials at this time, we put it in another anonymous page:
> > >
> > > https://anonymous.4open.science/w/supp-CamTrol-anonymous-5509/
> > >
> > > The anonymous page is the best way we consider that could exhibit these video results at this time. However, this page may suffer from instability (that's the reason why we put all things in our supplementary materials), and it might take about 5-15 minutes to back to normal if it receives too many requests in a short time.
> > >
> > > There are two main observations from these dynamic comparisons: 1) CogVideoX itself has rather static results even running without our method; 2) Our method does not have obvious declined dynamics compared with the original output of CogVideoX, and preserves well the dynamics in original videos.
> > >
> > > We ourselves consider CogVideoX currently the best open-source video generation model in the community. And CogVideoX still has limitations in generating videos with high dynamic moves.
> > >
> > > If our previous supplements of user study could help address the issues of visual quality, or the concrete examples of novel results could help solve problems of novelty, or the dynamic comparisons we post in this comment could help clarify the dynamics of generated results, we would be more than grateful that reviewer 14Er could adjust the score based on these addressed issues. If reviewer 14Er still found the problems not solved, please let us know for further discussion.
> > >
> > > Best regards,
> > >
> > > Authors

---

### Official Review · Reviewer_syyx · 2024-10-23

**Soundness:** 2
**Presentation:** 3
**Contribution:** 2
**Rating:** 6
**Confidence:** 4

**Summary:**

This paper propose a training-free camera movement control for off-the-shelf video diffusion models. Firstly, the image is converted to 3D point cloud space using depth estimator, and the video frame is re-rendered using the camera motion trajectory. In this process, inpaint is used to repair the vacant part of the point cloud in the rendering result, and finally, generate videos using layout prior of noisy latents(inversion from the rendering result). The experiments and demo show good results.

**Strengths:**

The presentation of this paper is clear. The proposed method is reasonable, effectively generating videos with camera trajectories from a higher-dimensional space (3D point cloud). Notably, the proposed method is training-free, allowing for easy adaptation to new fundamental video generation models and related research.

**Weaknesses:**

1. While the overall pipeline is effective, it is relatively complex, including depth estimation, rerendering, 2D image inpainting, video fundamental model redrawing, and other processes. Additionally, the integration of these various components is not particularly seamless. This complexity can lead to reduced generation efficiency and may result in inconsistencies between the different stages, which could ultimately compromise the quality of the output.
2. The presentation of the experimental part was poor, and the comparison with the relevant work and the combination with the latest video base model were not seen in the demo web page, although they mentioned it in the paper.

**Questions:**

1. In the quantitative evaluation, could you clarify the differences in the experimental settings for MotionCtrl, CameraCtrl, and other related papers? I noticed that there does not appear to be a uniform experimental setup in the field of camera control video generation.
2. The paper states that it compares CamTrol with state-of-the-art works, including AnimateDiff, MotionCtrl, and CameraCtrl. While it mentions that "AnimateDiff is excluded from quantitative analysis due to its incapacity to handle complicated trajectories," I still have not seen a qualitative comparison with AnimateDiff. Additionally, could you consider comparing your work with other relevant studies, such as CamCo[1] and Vd3d[2]?
3. The current results are excellent in terms of trajectory control, but they do not satisfy me in terms of video quality. Given the availability of a superior fundamental video model (with larger resolution and longer duration), and considering the combination with CogVideoX presented in this paper, I would appreciate it if you could provide some generated samples or comparisons for a more thorough evaluation. I will adjust the final score based on these results.

[1] Xu, Dejia, et al. "CamCo: Camera-Controllable 3D-Consistent Image-to-Video Generation." arXiv preprint arXiv:2406.02509 (2024).
[2] Bahmani, Sherwin, et al. "Vd3d: Taming large video diffusion transformers for 3d camera control." arXiv preprint arXiv:2407.12781 (2024).

---

> ### Author Response · Authors · 2024-11-23
> **Rebuttal of CamTrol to Reviewer syyx**
>
> We appreciate reviewer for the meticulous review and insightful questions.
>
> We will answer the questions as follows:
>
> **Q1**: Can you clarify the details in quantitative evaluations? It seems there doesn't have a uniform experimental setup in camera control video genration.
>
> **A1**: Yes, there doesn't have a uniform experimental setting in this domain. In our experiments, we randomly sampled 500 videos from RealEstate10k dataset as reference, and use their prompt-trajectory pairs for generation. In the previous version of paper, we used the default video base model for each method, i.e., VideoCrafter for MotionCtrl and AnimateDiff for CameraCtrl. We've changed this setting by using SVD as base model for every method.
>
> SVD is originally trained on resolution of 576 × 1024, but SVD-based CameraCtrl only support 320 × 576. Since changing the original resolution leads to suboptimal generation quality, we use 576 × 1024 for MotionCtrl and CamTrol, then resize their outputs into 320 × 576 for calculating metrics. For all methods, the number of frames and the decoding size of SVD are set to 14. We use 25 steps for both inversion and generation processes. We updated the detailed setting for quanlitative comparisons in the revised versin of our paper, starting from Line 304 (the revised parts are highlighted in blue). We also updated the newest quantitative results in Table.1. We also provide the video comparisons in the supplementary materials, in an html named "rebuttal_1_qualitative_comparisons_new".
>
> **Q2**: Not any qualitative comparisons with AnimateDiff.
>
> **A2**: In the previous version of our paper, we have qualitative results of AnimateDiff in Fig.3. Since we unified the video base models and conduct re-evaluations after, this figure is also altered. But we kept AnimateDiff's comparison in the supplementary, which we uploaded along with the paper submission. For better comparison, we made it into an html file named "2. comparisons_old". The qualitative results of AnimateDiff are shown here.
>
> We exclude the comparisons with AnimateDiff in the revised paper because: 1) AnimateDiff itself is a video base model, and the quality of base model affect heavily the results of camera-control videos, we can't compare with it fairly when other methods applying SVD as base model; 2) AnimateDiff cannot handle complex trajectories, and the scale of its basic trajectory is unknown. Thus, even evaluated under basic trajectories, the motion accuracy cannot be calculated as there is no ground truth.
>
> **Q3**: Comparisons with Camco and Vd3d.
>
> **A4**: Both Camco and Vd3d are not open-sourced, I'm afraid we can't conduct comparisons with these methods.
>
> **Q4**: Generation results under superior video foundation models.
>
> **A4**: Yes. During the experiments, we find that combining with better foundation models, CamTrol can generate videos with more consistent frames and more rational dynamics. We provide results of CogVideoX-5b+CamTrol in the updated supplementary materials (In the paper we used CogVideoX-2b), with an html file named "rebuttal_3_CogVideoX_results". CogVideoX-5b is trained at 8 fps, 720x480 resolution with 49 frames. Note that we exhibit the raw outputs of the model, without any post-processing algorithms like super-resolution or frame interpolation. These results, compared with SVD, are more alike to real-world videos with camera movements.
>
> We've also tried the latest mochi model by genmo, but it seems its basic outputs are not ideal even without any camera control.
>
> Secondly, please let us explain about the weakness:
>
> **W1**: Different stages of pipeline may cause inconsistent, and ultimately compromise the output quality.
>
> **E1**: We presented in Fig. 6 that the failure of inpainting and point cloud reconstruction possesses little damage to the output quality. For diffusion models, the operation of adding and removing noise have a effect of purifying imperfect inputs, as both perfect and imperfect inputs share the nearly-gaussian latent distribution. Similar ideas are wildly used in previous papers (SDEdit [1], Ilvr[2]). Thus, the performance in previous stages (depth estimation, inpainting, point cloud reconstruction) are of highly tolerance in our method, since they do not affect ultimate generations but only serve as layout guidance which indicates perspective changes.
>
> **W2**: Experimental part was poor. Qualitative comparisons are not shown on demo page.
>
> **E2**: We updated the revised version of our paper with unified and detailed experimental settings. Please check them for more detailed experimental settings and analysis.
>
> The qualitative comparisons with state-of-the-art methods are attached in the supplementary materials, in parallel with the demo web page, we kept it in the revised supplementaries for further review.
>
> [1] Sdedit: Guided image synthesis and editing with stochastic differential equations
>
> [2] Ilvr: Conditioning method for denoising diffusion probabilistic models

---

> > ### Comment · Reviewer_syyx · 2024-11-25
> >
> > Thanks to the author's reply, after reviewing the revised manuscript and the effect of combining CamTrol with the existing SOTA video generation model, I decided to improve my score.

---

> ### Author Response · Authors · 2024-11-25
> **Reply to Reviewer syyx**
>
> We thank reviewer syyx for improving the score, we really appreciate it.
>
> Best regards,
>
> Authors

---

### Official Review · Reviewer_9UGe · 2024-10-31

**Soundness:** 4
**Presentation:** 3
**Contribution:** 3
**Rating:** 6
**Confidence:** 4

**Summary:**

This paper proposes a framework to control the camera viewpoints of the video generation process. This frame work is training free, utilizing some existing pre-trained models, like depth estimation model, and the depth coefficient optimization model. It can be used on many different base video generators. Given a real or T2I-generated images, this frame first generate a series of images according to the camera trajectories, with the point clouds as the middle 3d representation. Then, these images are noised to a certain degree. After that, these noised latents are denoised with the video generator, getting the videos with desired camera trajectory.

**Strengths:**

1. The framework is totally training free, combining some pre-trained models.
2. The written is easy to follow.
3. Visualization results demonstrate the effectiveness of each part in this pipeline.

**Weaknesses:**

1. In Table.1 , some quantitative comparison between the SVD and CamTrol+SVD is lacked, using the FVD, FID, IS, and CLIP-SIM, metrics. This comparison can reflect the impact of using proposed pipeline on the pretrained video generators.
2. In the demo video, there are some obvious consistency or unreasonable on objects in generated videos. One possible reason is the in painting model cannot handle some situation well, for example, when the camera motion is large, leading much holes in the images, the inpainting model cannot get enough context to generate reasonable contents for the holes.
3. Continue with the second point, this pipeline may have some difficulty in properly use the pre-trained video generator to generate consistent and reasonable videos.

**Questions:**

Besides the first point in the weakness, I have the following questions.
1. In Table 1, what are the base video generator for MotionCtrl and CameraCtrl? Can you use the same base video generator for Motion, CameraCtrl, and CamTrol, like SVD to quantitatively evaluate them?
2. In the camera motion modeling step, from the same first image, it seems that the CamTrol can generate different image sequences, according to the camera trajectories. If seeding these image sequence into the second step (layout prior of noise). Can we get some videos with similar appearance but different camera trajectories? Can you have some comparisons with the CVD (Collaborative Video Diffusion: Consistent Multi-video Generation with Camera Control) method?
3. How to find the best trade-off of t_0 for each videos?

---

> ### Author Response · Authors · 2024-11-23
> **Rebuttal of CamTrol to Reviewer 9UGe**
>
> Thanks for your meticulous review, and for your positive comments on our work.
>
> We'll answer the questions above:
>
> **Q1**: Base models in Table.1.
>
> **A1**: We used the default model of each method, i.e., VideoCrafter for MotionCtrl and AnimateDiff for CameraCtrl. We've corrected this unsuitable setting and re-evaluate all methods based on the same SVD models. The results are updated in Table.1 in the revised version of our paper (the revised parts are highlighted in blue). We also provide video comparisons in the updated supplementary materials, in an html file named "rebuttal_1_qualitative_comparisons_new".
>
> **Table. 1 Quantitative Comparisons**
> | |FVD|FID|IS|CLIP-SIM|ATE|RPE-T|RPE-R|
> |-|-|-|-|-|-|-|-|
> |SVD|1107.93|68.51|7.21|0.3095|4.23|1.79|0.021|
> |MotionCtrl+SVD|810.59|69.03|7.17|0.3076|4.19|1.17|0.012
> |CameraCtrl+SVD|951.80|67.59|6.82|0.3138|4.22|1.07|0.013|
> |CamTrol+SVD|778.46|68.06|7.05|0.3110|4.17|1.07|0.010|
> |Reference|||||3.60|0.89|0.008
>
>
> **Q2**: Can we get some videos with similar appearance but with different trajectories? Comparison with CVD.
>
> **A2**: Yes! This is a great application of our method to generate multi-trajectory videos for one scene. We've added this into the revised version of our paper, in Fig.9 and Line 517-521. We also provide some video results for multi-trajectory generation in the supplementary materials. For better visualization, we made it into an html file named "rebuttal_2_CVD-like_results". You can click it to see some CVD-like results.
>
> However, as CVD is not open-sourced yet, I'm afraid we cannot conduct comparisons with it.
>
> **Q3**: How find the best trade-off of t_0?
>
> **A3**: In previous works (SdEdit [1], DiffusionCLIP [2]), similar early-stopped inversion steps are often manually set. Since inference of image/video generation could be done within minutes, it is fast and convenient to produce several outcomes and compare them. We offered some recommended choices of t_0 for different kinds of moves in Line 269, they are empirical findings, but there’s one rule that can be referred: Motions with drastic moves prefer larger t_0, as larger t_0 leads to less noise thus more relevance between adjacent frames of latents. For moderate moves, latents are relevant to each other in large spatial regions, which is easier for video model to recognize, thus t_0 of moderate camera motion doesn’t have to be very large.
>
> Secondly, please let us explain about the weakness:
>
> **W1**: This weakness refers to the same issue with Q1.
>
> **W2**: Inpainting models cannot handle some situations well, leading to inconsistency or unreasonable objects.
>
> **E2**: Inpainting models can fail in some situations, but these failure doesn't have severe impact on the generated videos. We present this point in Fig. 6, to show that the imperfections of inpainting possess little damage to the output. For diffusion models, the operation of adding and removing noise have a effect of purifying imperfect inputs, as both perfect and imperfect inputs share the nearly-gaussian latent distribution. This makes our method highly tolerant of the inpainting model's performance.
>
>
> **W3**: CamTrol may have some difficulties in properly using pretrained video models to generate consistent and reasonable videos.
>
> **E3**: Utilizing pretrained video models to generate camera-control videos relies heavily on the performance of pretrained models. For most open-sourced video models, their ability for generating consistent and reasonable scenes is not sufficiently enough, even without any camera control (e.g. SVD). During the experiment, we find that applying CamTrol on better video generation models lead to more consistent and reasonable videos. We put the results of combining CamTrol with CogVideoX-5b in the updated supplementary materials, in an html file named "rebuttal_3_CogVideoX_results". Using these models, CamTrol can generate more videos with more reasonable content, consistent scenes and dynamic movements.
>
> [1] Sdedit: Guided image synthesis and editing with stochastic differential equations
>
> [2] Diffusionclip: Text-guided diffusion models for robust image manipulation

---

> > ### Comment · Reviewer_9UGe · 2024-11-25
> >
> > I thank the review's time to do extra evaluations to address my concern on the unfair comparison. And the newly added results to show the application of CamTrol. I will keep my initial rating.

---

> > > ### Author Response · Authors · 2024-11-25
> > > **Reply to Reviewer 9UGe**
> > >
> > > We thank reviewer 9UGe for the positive comments of our work.
> > >
> > > Best regards,
> > >
> > > Authors

---

### Official Review · Reviewer_WcRk · 2024-11-04

**Soundness:** 2
**Presentation:** 2
**Contribution:** 2
**Rating:** 6
**Confidence:** 5

**Summary:**

This work proposes a training-free solution for camera control in video generation, requiring no supervised fine-tuning. A given image is first projected into 3D point cloud space and a sequence of views are then rendered given the camera trajectory. The latter serves as a structure condition for generation through inversion.

**Strengths:**

The writing is pretty clear, starting from the observation that camera movement could be regarded as one latent layout rearrangement.
The two-stage framework is well-presented and easy to understand.
The ablation study is comprehensive for multiple designs of proposed methods.

**Weaknesses:**

There are multiple major concerns:
- The novelty of this pipeline is limited which combines point cloud reconstruction and inversion. For example, Infinite Nature [a] also uses a RGBD image together with rendering to generate novel views and then refine.
- The claim for training-free could be further clarified since depth coefficient optimization is also adopted (as mentioned at L190). Although the base model is not tuned, this optimization could be empirical and time-comsuming, preventing it from the practical applications.
- As stated at L210, video should be encouraged with more dynamics, which is quite important. Yet, we do not find any design/modification for this.
- From one side, inversion provides the trade-off between fidelity and diversity (L261). From the other, it is challenging to choose which timestep should be used for the generation with given camera conditions.
- How the proposed method tackle text-to-video generation tasks? Fig.1 suggests that an input image is always given.


[a] Infinite Nature: Perpetual View Generation of Natural Scenes from a Single Image

**Questions:**

Please check the weaknesses

---

> ### Author Response · Authors · 2024-11-23
> **Rebuttal of CamTrol to Reviewer WcRk**
>
> Thanks for your review and for your questions, we will answer them one by one:
>
> **Q1**: Lack of novelty.
>
> **A1**: Infinite Nature uses camera-annotated dataset to train a network for generating novel views, and its dataset only contains images of aerial coastlines.
>
> Our method doesn't need any additional network, and doesn't need training on limited datasets that have camera pose labels. More importantly, methods trained on such datasets, e.g., MotionCtrl, CameraCtrl and Infinite Nature, can only handle camera control of certain scenes. For Infinite Nature, it can only deal with images depicting aerial coastlines. Nor can they handle the generation of dynamic content, as these datasets only contain static scenes.
>
> **Q2**: Optimization is time-comsuming, preventing it from practical applications.
>
> **A2**: We tested the time consumption of our method. The optimzation process takes about 0.42s for one frame at 576x320. For a video with 14 frames, the optimization process only takes about 6s. We offer more detailed computational analysis in the revised version of our paper in Table 2 (the revised parts are highlighted in blue). Our method saves about 10-20 GB GPU memory than other finetuned method under the same circumstances, and can be applied on a single RTX 3090 GPU.
>
> **Table2. Computational Analysis**
> ||SVD|MotionCtrl+SVD|CameraCtrl+SVD|CamTrol+SVD|
> |-|-|-|-|-|
> |GPU memory (max)/MB|11542|31702|26208|11542|
> |Time (s)  - preprocess |-|-|-|56|
> |Time (s) -  inference|11|32|42|8|
>
> **Q4**: No design for encouraging dynamics in video.
>
> **A4**: In Line 261 - 269, we discussed how to encourage generated videos with more dynamics or more alignment with input trajectories. In Fig.7, we illustrated how the choiice of t_0 effect the output in dynamics and motion alignment. We also updated our paper with a more detailed quantitative evaluation of different t_0s.
>
> **Q5**:  t_0 is challenging to choose.
>
>
> **A5**: Similar early-stopped inversion steps are often manually set in previous works (SdEdit [1], DiffusionCLIP [2]). Unique value of t_0 may not be easily defined, but multiple choices provide users with more flexibility to decide whether they want videos with more dynamics or align better with given trajectories. Inference of video generation can be conducted within minutes, and it is fast to compare results with different t_0. We offered some empirical choices for t_0 in Line 269.
>
> **Q6**: How it tackle text-to-video tasks?
>
> **A6**: In Line 163, we explained that image could be created by image generators like Stable Diffusion. In Line 751, we clarified the image generation models we use for handling text input.
>
> We considered adding this into Fig.1, but excluded it to keep the diagram’s layout intact.
>
> [1] Sdedit: Guided image synthesis and editing with stochastic differential equations
>
> [2] Diffusionclip: Text-guided diffusion models for robust image manipulation

---

> > ### Comment · Reviewer_WcRk · 2024-11-26
> >
> > Thanks for your rebuttal that clearly addresses my concerns of time cost, and text-to-video tasks. But I remain worried about how the proposed method tackles the dynamic videos. Can we show more results of dynamic videos or any supports?

---

> ### Author Response · Authors · 2024-11-26
> **Second rebuttal to Reviewer WcRk**
>
> We thank reviewer WcRk for the clarification and for the response.
>
> Yes, for generating dynamic results, we 1) added some comparisons in Fig.4 of the revised version of paper, to demonstrate our method's ability in tackling more generic, dynamic scenes, while other finetuned methods fail to do so; 2) added more video examples in supplementary materials, html file named "second_rebuttal_dynamic_comparisons", to exhibit how different methods perform on generating videos that contain dynamic content, basd on SVD. Note that the open-source SVD itself does not perform very well in generating dynamic content (most of its examples are videos with static scenes, which could be referred to through https://huggingface.co/stabilityai/stable-video-diffusion-img2vid), thus all four methods based on SVD (including SVD itself) struggle to produce highly dynamic results, but still our method performs better than others in generating videos with dynamic content. 3) We added more generating samples of our method based on SOTA video generative model CogVideo-X in the supplementary materials, in the html file named "rebuttal_3_CogVideoX_results", where the results have more dynamics than SVD while align well to the input camera motion. 4) We provide even more dynamic videos generated by CamTrol in the html file "second_rebuttal_more_dynamics", supplementary materials, which have considerable movements of the generated content. There are also high-dynamic videos presented in the anonymous demo page (which can also be referred to in the supplementary materials, file "1. anonymous_demo_page"). These files contain many camera-moving videos with quite high dynamics in their generated content, and can stand as proofs that our method is capable of handling dynamic video generation with camera control.

---

> > ### Comment · Reviewer_WcRk · 2024-12-02
> >
> > Thanks for your update that further addresses my concerns of dynamics. I this raise my score to 6.

---

> > > ### Author Response · Authors · 2024-12-02
> > > **Reply to Reviewer WcRk**
> > >
> > > We truly thank reviewer WcRk for raising the score above acceptance threshold, it means a lot to our work.
> > >
> > > Best regards,
> > >
> > > Authors

---

### Official Review · Reviewer_4R7b · 2024-11-08

**Soundness:** 2
**Presentation:** 3
**Contribution:** 3
**Rating:** 6
**Confidence:** 5

**Summary:**

The paper introduces a method named CamTrol to enable controllable camera movements for video generation using off-the-shelf video diffusion models. The innovation lies in its training-free approach—no need for additional supervised or self-supervised training. The method employs a two-stage process:
1. 3D Camera Motion Modeling: It use a single view image as input to regressive a 3D point cloud, then gradually rendering, refining the 3D point cloud. The rendering process adopt the camera poses to simulate camera motion. The generated image sequence is later used to guide the perspective and layout in generated video sequences.
2. Noise Prior Guidance: It utilizes the “layout prior” of noisy latents, first adding noise of the image sequence, injecting the camera motion prior to the noisy latents. Then, providing the noisy latents into a pretrained video generation model to generation a reasonable and camera viewpoint alignment video.

**Strengths:**

1. CamTrol is a training-free solution, it does not need additional training, making it computationally efficient and easy to integrate with existing video diffusion models.
2. It investigate the feasibility to adopt the “noise prior of latent” technique to control the camera viewpoint without direct supervision.
3. CamTrol outperforms competing methods in both perceptual quality and motion alignment, particularly in complex camera movements, as demonstrated in both quantitative and qualitative experiments.

**Weaknesses:**

1. Since the whole pipeline is complex (depth estimation -> point cloud lifting -> rendering -> inpainting -> depth coefficient optimization), CamTrol’s quality is vulnerable. A problem at each of the current steps will have a bad effect on the next step. For example, if the depth estimation model cannot give the precise results, the structure of the entire scene may be strange when viewed from the different perspectives. If the inpainting model is not good enough, the generated videos may have much inconsistency.
2. For different videos (and different camera trajectories), the best timestep t_0 may be inconsistent. It is hard to design a rule to choose the best t_0.
3. The inpainting model is designed for single images, which may lead to inconsistent content and difficulty in generating reasonable new content. Additionally, the noisy latent not only provides layout priors but also carries some semantic information, which is unfavorable for the video generation model when creating new content. This makes CamTrol potentially unsuitable for generating long videos.

**Questions:**

1. What is the evaluation dataset for the Table.1 ? If the references are come from the RealEstate10K dataset (Line 322), it is strange that the Video quality metrics of MotionCtrl and CameraCtrl are worse than the CamTrol, since both MotionCtrl and CameraCtrl are fine-tuned on the RealEstate10K dataset.
2. What is the base video generation models of MotionCtrl and CameraCtrl in Table.1 ? If the base video generators are not SVD, this comparison in Table.1 is unfair. Besides, can you provide the Video Quality of the vanilla SVD model for a reference?
3. Since the ParticleSFM may fail sometimes, a natural question is: how accurate is the camera computed ParticleSFM on the ground truth videos? It would be good to have some reference number on the ground truth dataset to show the lower bounds of Motion Accuracy metrics in Table.1
4. In some cases of the provides webpage,  with the same camera trajectory (especially on complex trajectories), different videos do not have the same camera movement. For example, the complex trajectory II, the first video shows downward at the end of the video, the third video shows upward, the last video ignores the last camera movement. However, MotionCtrl and CameraCtrl do not have this kind of inconsistency. Can the author provide some quantitative results on complex camera trajectories?

---

> ### Author Response · Authors · 2024-11-23
> **Rebuttal of CamTrol to Reviewer 4R7b**
>
> Thanks for your careful review and meticulous comments for our work.
>
> We will answer the questions mentioned above:
>
> **Q1**: Evaluation Details.
>
> **A1**: Yes, the evaluation dataset in Table.1 is RealEstate10k. We used text prompt from MotionCtrl for video generation, these prompts are not extracted from RealEstate10k and might cause domain gap. We assume this is the reason why our method surpasses them even they've been finetuned on RealEstate10k.
>
> **Q2**: Base model of MotionCtrl and CameraCtrl in Table.1
>
> **A2**: We used their default setting of base model, i.e., VideoCrafter for MotionCtrl and AnimateDiff for CameraCtrl. We've corrected this setting and unified the video base model to SVD for all methods. Following your advice, we also include the video quality of vanilla SVD as a reference. The results are shown in the table below, we've also updated it into the revised version of our paper (the revised parts are highlighted in blue). We also provide video comparisons in the updated supplementary materials, in an html file named "rebuttal_1_qualitative_comparisons_new".
>
> **Table. 1 Quantitative Comparisons**
> | |FVD|FID|IS|CLIP-SIM|ATE|RPE-T|RPE-R|
> |-|-|-|-|-|-|-|-|
> |SVD|1107.93|68.51|7.21|0.3095|4.23|1.79|0.021|
> |MotionCtrl+SVD|810.59|69.03|7.17|0.3076|4.19|1.17|0.012
> |CameraCtrl+SVD|951.80|67.59|6.82|0.3138|4.22|1.07|0.013|
> |CamTrol+SVD|778.46|68.06|7.05|0.3110|4.17|1.07|0.010|
> |Reference|||||3.60|0.89|0.008
>
>
> **Q3**: As ParticleSFM may fail, motion metrics shoule be provided with lower bounds.
>
> **A3**: Yes, we should include the scores evaluated on the ground truth videos as lower bounds of these motion metrics. We've integrated these results into the table above, and also in the revised paper.
>
> **Q4**: Same trajectory may cause different moves, and can author provide quantitative comparisons on complex trajectories?
>
> **A4**: The results in Table.1 are tested under complex trajectories. We randomly sampled 500 trajectories from RealEstate10k. The reason that same trajectories may cause different moves is we used different t_0. If we set t_0 to bigger value, all results will have the same movements that align to the input trajectory, but this will also lead to loss of video dynamics. Setting smaller t_0 will allow the generated videos with more dynamic, while generally follow the input camera motions. As MotionCtrl and CameraCtrl are finetuned on camera-annotated dataset, they have less output inconsistency for the same trajectory, but this only works on static, real estate-like scenes with non-dynamic content.
>
> Secondly, please let us explain shortly about the wearkness:
>
> **W1**: Each of the current step will have a bad effect on the next step.
>
> **E1**: The operation of adding and removing noise make diffusion model capable to deal with imperfect inputs, as both perfect and imperfect inputs share the nearly-gaussian latent distribution. Similar ideas are wildly used in previous papers (SDEdit [1], Ilvr [2]). Thus, the performance of depth estimation and inpainting are of highly tolerance in our method, since they only serve as layout guidance which indicates perspective changes. We presented some examples in Fig.5 to show that their inferior performance possesses little damage to the generation.
>
> **W2**: Different videos may have different t_0.
>
> **E1**:  Yes, similar early-stopped inversion steps are often manually set in previous works (SdEdit [1], DiffusionCLIP [3]). We offer some empirical choices for t_0 in Line 269 of the paper.
>
> **W3**: Inpainting may lead to inconsistency.
>
> **E3**: We carefully designed the inpainting process and depth optimization algorithm to improve consistency between adjacent views, we present this part in  Line 184 – Line 190 of the paper.
>
> **W4**: Latents can carry unfavorable semantics for video generation, making this method unsuitable for generating long videos.
>
> **E4**: Semantic and layout information are attached, it is the layout of semantic information in noisy latents that help diffusion model to identify what to represent in certain positions and how these content moves. We majorly rely on inpainting model instead of video model to complete scenes and generate new contents. After inversion, the semantic information of new content will be encoded into noisy latents. This process can be repeatedly performed regardless of number of frames.
>
> [1] Sdedit: Guided image synthesis and editing with stochastic differential equations
>
> [2] Ilvr: Conditioning method for denoising diffusion probabilistic models
>
> [3] Diffusionclip: Text-guided diffusion models for robust image manipulation

---

> ### Comment · Reviewer_4R7b · 2024-11-27
> **Comment**
>
> The author's rebuttal has addressed my main concerns; however, I still find the proposed pipeline is complex and lacking scalability. I tend to recommend acceptance, but with a borderline rating.

---

> ### Author Response · Authors · 2024-11-27
> **Reply to Reviewer 4R7b**
>
> We thank reviewer 4R7b for the positive review of our work.
>
> For complexity, the proposed pipeline doesn't require extra carefully-designed network; for scalability, the proposed pipeline can be generalized to most video diffusion models without finetuning, and can inherit scalability from base models (e.g., scene diversity, generation capability). We assume the proposed pipeline, in some cases, has advantages in these two terms, compared with finetuned methods.
>
> Best regards,
>
> Authors

---

### Author Response · Authors · 2024-12-03

We would like to thank all reviewers for their time and effort in the review process. We appreciate that the reviewers recognized the value of our work and found it "well-presented and clear" (WcRk, 9UGe, syyx), "easy to adapt" (4R7b, 9UGe, syyx), and our experiments "effective and comprehensive" (WcRk, 9UGe). We also thank the valuable and insightful comments from all reviewers that help us improve the paper, and we are happy to see our rebuttal has addressed their concerns (4R7b, WcRk, 9UGe, syyx, 14Er).

Following the suggestions and comments of reviews, we have revised the manuscript and marked the revisions in blue, listed as below:

+ We have re-conducted the quantitative comparisons based on the same video models. (Table 1, from Reviewer 4R7b, 9UGe)
+ To make the experiments more convincing, we have added quantitative comparisons with SVD, and tested the lower bound of motion metrics using ground truth trajectories. (Table 1, from Reviewer 4R7b)
+ We have added more detailed and clarified experimental setting in the evaluation part. (Section 4.1, from Reviewer syyx)
+ We have conducted computation analysis of time consumption and GPU memory for different methods. (Table 2 and Section 4.2, from Reviewer WcRk, 14Er)
+ We have added generalizability and dynamic comparisons of different methods. (Fig.4, from Reviewer 14Er)
+ We have added the quantitative ablation on t_0. (Reviewer 14Er)
+ Under the advice of Reviewer 9UGe, we also have explored a new application of our method, which is generating multi-trajectory videos with the same scenes. These results are shown in Fig.9 of the main paper as well as in the supplementary materials. (Fig.9 and supplementary materials, from Reviewer 9UGe)
+ We have tried our method on more SOTA video models, i.e., CogVideoX, and produced many high-quality, dynamic videos with camera control. We put these contents in the supplementary materials. (Supplementary materials, from Reviewer syyx, 14Er, WcRk)

---

### Meta-Review · Area_Chair_PuLH · 2024-12-20

**Metareview:**

The paper uses pre-trained video diffusion models to introduce a training-free method for controllable camera movements in video generation. By leveraging 3D point cloud modeling, camera motion simulation, and layout priors in noisy latent, it achieves flexible camera control without requiring additional datasets or fine-tuning, outperforming prior methods.

Agreed by most reviewers, CamTrol's strengths include (1) its training-free approach, which is flexible and adaptable for use with different foundation models; (2) the innovative use of noise priors in latent space to control camera motion without supervision; and (3) strong performance in visual quality and motion alignment, particularly for complex camera movements, as validated through experiments and ablation studies.

As pointed out by several reviewers, the limitations include (1) its complex multi-stage pipeline; (2) inconsistencies in scene and camera motion, with unintended changes or inaccuracies in trajectories; (3) the inpainting model’s inability to handle large camera motions, leading to unreasonable content generation; (4) significant computational overhead from depth coefficient optimization, and (5) missing quantitative comparisons in key experiments. The authors addressed the concerns about optimization overhead and quantitative comparisons, and partially addressed issues related to scene consistency and camera motion. Pipeline complexity and inpainting model limitations are not adequately discussed, but most reviewers were happy with the responses.

**Additional Comments On Reviewer Discussion:**

While the authors have addressed most of the major concerns and provided a comprehensive summary, Reviewer `14Er` still rates the paper below the borderline. The primary issue raised is the visual quality of the results. The reviewer contends that, given the paper's limited novelty and its focus on engineering contributions, higher video quality is expected.

Despite Reviewer `14Er`'s concerns, the ACs value the paper's technical contributions and flexibility. CamTrol's training-free approach and seamless integration with pre-trained models make it highly practical and impactful, justifying its acceptance.

---

### Decision · Program_Chairs · 2025-01-22

Accept (Poster)